# Position: Future Research and Challenges Remain Towards AI for Software Engineering

**Alex Gu** [1]   **Naman Jain** [2] [*]   **Wen-Ding Li** [3] [*]   **Manish Shetty** [2] [*]   **Kevin Ellis** [3]   **Koushik Sen** [2]
**Armando Solar-Lezama** [1]

## Abstract

AI for software engineering has made remarkable progress, becoming a notable success within generative AI. Despite this, there are still many challenges that need to be addressed before automated software engineering reaches its full potential. In this paper, our goal is threefold. First, we provide a taxonomy of measures and tasks to categorize work towards AI software engineering. Second, we outline key bottlenecks permeating today's approaches. Finally, we call for large open-source community efforts and lay out a collection of promising research directions to address these challenges, hoping that we can all come together to advance and shape the future of AI for code.

## 1. Introduction

AI for software engineering has made remarkable progress recently, becoming a notable success within generative AI. Despite this, there are still many challenges that need to be addressed before automated software engineering reaches its full potential. With additional efforts, it should be possible to reach high levels of automation where humans can focus on the critical decisions of what to build and how to balance difficult tradeoffs while most routine development effort is automated away. Reaching this level of automation, however, will require substantial research and engineering efforts across academia and industry. **This paper provides an opinionated view of the tasks, challenges, and promising directions towards achieving this goal.**

Many existing surveys overlap with the topics that are discussed in this paper, such as AI for programming assistants (Liang et al., 2024; Sergeyuk et al., 2025), LLMs for software testing (Wang et al., 2024c), using LLMs in low-resource and domain-specific languages (Joel et al., 2024),

automated program repair (Zhang et al., 2023c) focus on automated program repair, and formal mathematical reasoning (Yang et al., 2024d). In addition, many papers discuss the current state, challenges, and future of AI for software engineering (Fan et al., 2023; Ozkaya, 2023; Wong et al., 2023; Zheng et al., 2023; Hou et al., 2024; Jin et al., 2024; Wan et al., 2024b; Roychoudhury et al., 2025a). Our work draws inspiration from them, and we recommend that the reader consult with them for alternative perspectives.

In this paper, our goal is threefold. In Sec. 2, we provide a structured taxonomy of concrete tasks in AI for software engineering. In particular, we emphasize that there are many other tasks in SWE beyond code generation and code completion, encouraging research in these areas. Moving forward to Sec. 3, we highlight four main challenges that today's models face, each cross-cutting and applicable to several tasks. In Sec. 4, we call for large open-source community efforts in the field and lay out a collection of promising research directions to address these challenges. The main text contains a summary of highlights, with a full version in the Appendix. We hope that after reading our paper, the reader can appreciate the progress AI for code has made, understand the shortcomings of today's state-of-the-art models, and come together to advance and shape the future of the field.

## 2. Tasks in AI Software Engineering

We first provide a taxonomy of tasks in AI software engineering. To provide a structured way to consider each task, we define three measures that apply across them: scope, logical complexity, and level of human intervention. To achieve an AI software engineer, we strive for AI to be capable across the board for all three measures.

**Scope Measure**: We define three levels of scope, the extent of changes that the AI makes to the codebase. *Function-level* scope refers to single, self-contained functions such as in HumanEval (Chen et al., 2021a) and MBPP (Austin et al., 2021). *Self-contained unit* scope goes beyond singular functions and to larger chunks of code such as entire files and classes. *Project-level* scope refers to larger codebases

[*]Equal contribution  [1]MIT [2]UC Berkeley [3]Cornell. Correspondence to: Alex Gu <gua@mit.edu>.

*Proceedings of the 42nd International Conference on Machine Learning*, Vancouver, Canada. PMLR 267, 2025. Copyright 2025 by the author(s).

such as entire repositories, such as in Commit0 (Zhao et al., 2024) and SWE-Bench (Jimenez et al., 2024).

**Logical Complexity Measure**: Tasks require a wide range of reasoning abilities when it comes to devising algorithms to solve them. *Low logical complexity* tasks require little to no reasoning, such as writing CRUD (create, read, update, delete) applications or using APIs. *Medium logical complexity* tasks include most LeetCode problems, finding inputs to fuzz simple programs, and reasoning about execution behavior of multithreaded programs. *High logical complexity* tasks require meticulous and challenging levels of algorithmic and logical reasoning, either because the algorithm is complex or because the problem requires clever thinking and insights. This includes difficult competition programming problems, writing large thread-safe concurrent programs, cracking cryptographic ciphers, and solving SMT-like problems. Many popular coding benchmarks are function-level, medium-high logical complexity, such as APPS (Hendrycks et al., 2021), CodeContests (Li et al., 2022a), and LiveCodeBench (Jain et al., 2024b).

**Level of Human Intervention Measure**: AI coding is a collaborative task. We follow the autonomy taxonomy outlined in Morris et al. (2023) to define three levels of human intervention. *Low autonomy* is when the human has full control over the task and uses AI to automate simple sub-tasks. *Medium autonomy* is when the human and AI contribute a similar amount, with interactive coordination of goals and tasks. Here, the human and AI might both suggest refactorings and optimizations during the development cycle. *High autonomy* is when AI drives the interaction and tasks, identifying required changes and the changing demands of the user. The AI would defer to the human only when needed or for a final check, write the code and tests autonomously. Next, we turn to the set of tasks that are reflective of the tasks and capabilities of a human software engineer.

### 2.1. Code Generation

Code generation is the task of generating code from a specification. In *code completion*, the specification takes the form of a preexisting code snippet, and the goal is to complete the snippet. There are two popular paradigms for code completion: *tab completion*, where the user can press the tab key to complete a block of code (e.g. GitHub Copilot), and *natural language to code*, where the specification is a natural language description with requirements such as the task description or input-output examples. Recently, AI-driven IDEs have blurred the lines between the two paradigms. With the ultimate goal of decreasing the burden of human programmers, they aim to automatically infer the user's intent from the code context and user behavior. However, when intent is vague, they allow users to specify desired functionality via chat interfaces.

### 2.2. Code Transformation

**Code Refactoring**: In code refactoring, the goal is to take a working implementation of a piece of software and rewrite parts of it while maintaining correctness. One challenge with this task is that success extends beyond functional correctness or metrics. Because it can often be unclear what level of abstraction refactorings should be done at, completing a refactoring at a high autonomy level is also difficult. These challenges are further compounded by the need to understand implicit trade-offs customized to specific codebases, respect conventions, and reason about the long-term maintenance implications of structural changes.

**Code Migration and Translation**: An incredibly resource-intensive task is migrating large amounts of code while preserving all the original functionality. Such high-value migrations present opportunities for AI-assisted automation to reduce cost and manual effort. Code migration often involves changes across many files and systems with complex transformations. A special case of code migration is code translation (transpilation): rewriting code from a source language to a target language. In industry, this task can be motivated by several reasons such as security and scalability concerns in legacy languages or avoiding technical debt. Due to the safety-critical and cross-system nature of many migrations, this task often requires substantial human oversight in practice and cannot be done fully autonomously.

**Code Optimization**: Transforming programs to improve performance characteristics while maintaining functional correctness is a critical software task. Optimizing real-world systems poses significant challenges, as performance bottlenecks must be identified and new algorithms to mitigate them must be proposed. Code optimization often has a large search and solution space with competing objectives like speed, memory efficiency, and readability, for example when optimizing kernel code at the PTX level for GPU-based AI model optimization (Zhao et al., 2025; Ouyang et al., 2025).

### 2.3. Software Testing and Program Analysis

**Software Testing**: Software testing is a practical approach to prevent bugs, both during development and production. There are several popular approaches to software testing, some short-term and others longer-term. For example, *Unit testing* refers to using input-output style tests that exercise the functionality of a piece of code and *property-based testing* is based on formal specifications and relies on specifying test cases that ensure that known properties of the code hold. The goal of software testing is to design tests that can surface bugs reliably. This is evaluated through metrics such as code coverage–how much of the source code is executed when the test suite is run. While practical, software testing faces challenges such as the scalability limits of traditional

tools and the difficulty of manually designing tests with good coverage.

**Program Analysis**: While testing catches bugs, the most challenging software issues are security vulnerabilities and zero-day exploits, from memory corruption to privilege escalation. This requires a deep program understanding, that testing/fuzzing often misses. For instance, a *zero-day* is a vulnerability unknown to the software developers that is found by an attacker, and there is no patch available from the vendor. In such cases, the only practical approach is offensive security research, manual source code audits, and root cause analysis of prior vulnerabilities to harden codebases.

## 2.4. Software Maintenance

**Code Documentation and Summarization**: To ensure maintainability, readability, and ease of collaboration, code must be well documented. Good documentation needs to be written cleanly and crisply, describing what the function does and how the function works. It must also anticipate and address any misunderstandings that a programmer might have, such as potential side effects or special cases. Humans often see documentation as a chore and neglect it, leading to code and documentation frequently being out of sync. This has led to the concept of "self-documenting code", code that clearly conveys its purpose.

**Pull Request (PR) Review**: Reviewing pull requests is an integral aspect of the software development cycle. While the most essential requirement for PRs is that a new feature is implemented correctly, other important considerations include checking whether the repository's style conventions are satisfied, ensuring that the PR does not introduce any new bugs, verifying that program invariants and guarantees still hold, and inspecting whether tests are robust.

**Code Understanding, Navigation, and Question Answering**: When encountering a codebase for the first time, developers often find it challenging to understand and develop a good mental model of the code. This can be due to many reasons: too many wrapper functions, excessive error-handling boilerplate, deep call stacks, or poor code cleanliness. One important challenge in code understanding is *code navigation*: finding where relevant functionality is implemented. Doing this well requires understanding the high-level layout of where every functionality lies in the codebase and the low-level understanding of which helper functions are used to implement each functionality.

Another challenge is *code question answering*: answering complex questions about a codebase, which requires sophisticated code understanding and reasoning abilities. Models should not hallucinate or give incorrect information that skews a developer's mental model of the code. Beyond other tasks mentioned in this section, developers might commonly ask questions related to data flow (when and where data structures get mutated), code functionality (whether there are any side effects), performance characteristics (determining the runtime and memory complexity of a function), or error handling (whether certain corner cases are handled).

## 2.5. Scaffolding and Meta-Code

For a software system to work, the core logic must be written well, but that is not enough. Many infrastructural aspects must be in place to support the software. We group these into two main categories: *scaffolding* and *meta-code*. We define *scaffolding* as a task outside of the code that must be done to get the software running properly. Examples of scaffolding include setting up Google authentication, subscribing to APIs, managing file storage, and generating API tokens. In contrast, we define *meta-code* to be code that is important to make the system work but does not actually participate in the execution of its main logic. Examples of meta-code include test harnesses, files, CI/CD code, Makefiles, Dockerfiles, sandbox databases, and preprocessors. Scaffolding and meta-code often are small in scope and have low logical complexity but can require a lot of domain-specific knowledge about the application, requiring human intervention.

## 2.6. Formal Verification

The task of formal verification involves generating checkable, mechanized proofs that can guarantee that a piece of code works as intended. Formal verification of software is necessary in mission-critical applications such as aircraft software, where it is crucial that code is correct with absolute certainty. Over the years, there have been countless programming languages designed specifically for formal verification. Some of the popular ones include TLA (Lamport, 1994), Coq (The Coq Development Team, 2024), Lean (De Moura et al., 2015), Dafny (Leino, 2010), Isabelle (Nipkow et al., 2002), and Verus (Lattuada et al., 2024). While formal verification tools have begun to see adoption in industry, they has not yet become mainstream because of these challenges. Code LLMs could greatly ease this burden and make it more feasible to verify code at larger scales, especially verifying properties requiring lower logical complexity.

## 3. Challenges

While the field of AI for code has made fruitful progress, cutting-edge AI still struggles with SWE tasks, especially at larger scopes and higher levels of logical complexity. Next, we discuss four high-level challenges in AI for code: data, scale, interaction, and measurement. These four challenges permeate across all of the tasks mentioned in the previous

section.

## 3.1. Data

**Low-Resource Languages and Specialized Libraries**: As we adapt code LLMs to individual codebases, generating correct code in out of distribution (OOD) scenarios becomes crucial. Much of software development in business contexts revolves around proprietary codebases, which is a distribution shift from the open-source code that dominates LLM training data (Ahmed et al., 2024a). These OOD scenarios include domain-specific languages (DSLs), custom internal libraries, low-resource APIs, and company-specific coding styles and conventions.

In low-resource languages, models have a weaker semantic understanding of the language. Due to the lack of training data in these OOD domains, models may struggle to write common primitives or piece together functionality coherently. On HumanEval, Qwen-2.5 has an accuracy of 83% in Python but 27% in D (Hui et al., 2024). In OOD scenarios, LLMs lack awareness of the libraries and functions available for use. In new codebases using custom libraries, many functions appear only a few times, providing limited training data for AI models to learn their usage. This scarcity can lead to overfitting, where models fail to recognize an effective use-case of these functions. Models also frequently hallucinate non-existent functions based on patterns that it infers.

**Library and API Version Updates**: Continual learning, the idea of training an AI system to take in new information continually, has been a long-standing challenge in AI and NLP (Wu et al., 2024; Wang et al., 2024d). In software engineering, codebases are continuously changing as new features are supported and awkward design patterns are reworked. While backwards compatibility is often prioritized in software design, it inevitably becomes broken as codebases evolve further. Therefore, programming libraries have version releases, each release supporting and deprecating features in the last version.

Even identifying which version of a library is being used can be quite difficult, because versioning information can be hidden deeply within a codebase. Sometimes, it can be found in comments or configuration files, but in the worst case, it must be inferred from the library calls being used. To make things worse, some code may be compatible across multiple versions, while other code will cause errors only in specific versions. On the other hand, writing code for a specific version can also be challenging, because it can be difficult for LLMs to implicitly keep track of which constructs and patterns are associated with each version. For example, when asking LLMs to write code in Lean 4, it often uses constructs from Lean 3, an older version that is much more prevalent across GitHub.

**High Logical Complexity and OOD Domains**: Some programming tasks are challenging for even the best human programmers, requiring approaches with a very high logical complexity. Examples of tasks that fall into this category include superoptimizing programs, discovering attacks for purportedly secure code, writing performant compilers, optimizing GPU kernels (Ouyang et al., 2025), and writing concurrent programs.

Because they are hard for humans, these tasks are very rarely in the training data of today's language models. They have unique, domain-specific, challenges that making generalizing from existing data difficult. For these problems, language models rely heavily on feedback-driven search algorithms (Mankowitz et al., 2023b), and it can be difficult to navigate the search space effectively.

## 3.2. Scale

**Large Scope and Long Contexts**: At the repository level, the tasks in Sec. 2 become significantly more difficult and require many steps. In code generation, user alignment can be an issue because there are many decision points and tradeoffs that can compound. In code refactoring, modifications will touch multiple parts of the codebase, and it can be tricky to keep the repository consistent. In code debugging, functions can be large and bugs can be nested deeply within stacks of function calls. Another issue with large scopes is large context lengths. Software engineering often requires dealing with very large codebases–for example, Google has repositories with over a billion lines of code (Potvin & Levenberg, 2016). As this is far too large for modern-day LLMs, choosing the correct context to include when using LLMs is important.

*Limits of Retrieval-Augmented Generation (RAG)*: Retrieval-based algorithms have been the predominant way to deal with long-context coding issues. First, the retriever retrieves relevant functions. Then, the generator leverages the retrieval to improve generation. While RAG has proven effective in many NLP tasks, the code domain provides new challenges for these methods.

*Retrieval*: In most NLP tasks, the retrieval step can be done relatively well because keywords that are in the query are often keywords that need to be retrieved. Unlike answering NL questions, writing code often requires drawing inspiration from code snippets that may be completely different syntactically. This can include programs with similar semantics, algorithms, or API calls, all of which potentially have very little in common when it comes to syntax. Previous work such as Ma et al. (2024); Utpala et al. (2023) have found that code in the same language is, on average, far closer in embedding space compared to semantically equivalent code.

*Generation*: In NLP tasks, the generation step often is a straightforward application of the retrieved information. In code, we are dealing with code reuse: piecing together relevant snippets of code in a precise and productive way to fit the current context. Each piece of retrieved content provides different information. This can include information about the language's syntax, documentation about the API, clues about the algorithm to be written, or examples of similar functionality being written.

**Long-Horizon Code Planning** When working on large projects, engineers often make complex decisions about how to design and structure the code to best support the various functionalities that will eventually be needed. To build a long-lasting software system, an engineer must know the potential paths that the system's evolution might take. This requires domain expertise and experience in how different code structures require different forms of extension.

*Designing Good Abstractions*: One instance of long-horizon code planning is choosing the right abstractions from the outset. An API designed with good abstractions will allow new features to be implemented seamlessly with minimal user overhead, while an API designed with poor abstractions may lead to excessive code duplication, refactoring, or even debugging. An example of this is *library learing*: finding the right APIs and libraries that can provide useful abstractions, leading to more reuse and intuitive interfaces. (Ellis et al., 2021; Stengel-Eskin et al., 2024; Bowers et al., 2023). While the traditional library learning literature has focused primarily on code reuse, a truly effective library must also prioritize ease of use and maintainability, as well as be robust and adaptable to future extensions.

**Modularity and Code Quality**: LLMs are trained and optimized primarily for code correctness with insufficient focus on other aspects of code like quality and maintainability. This is further exacerbated with large scale reinforcement learning being performed using test cases which can lead to unintended consequences regarding code quality, as correct but poor quality code is still often given a high reward. Empirically, it has been observed that LLM written solutions are often more complex than human-written counterparts. For example, Jain et al. (2024c) identified that LLMs prefer to repeat existing code instead of making use of abstractions in the existing code. One aspect of code quality is modularity, ensuring that code does not get duplicated too often. Here, Berlot-Attwell et al. (2024) identified that library or tool reuse is non-trivial for LLMs in coding and formal math domains.

**Effective Tool Usage**: Software engineering has witnessed the development of various open and proprietary tooling support for programming, debugging, analysis, and code management over time. For example, program analysis tools provide static and dynamic assurances on code correctness. Print statements and debuggers are used for dynamically analyzing and debugging programs at a fine-grained level. Beyond programming, such tools are richly integrated into the entire software development lifecycle, e.g., code navigation or search, reviewing code, CI testing.

While many efforts combine LLMs and agents with tools, they do not achieve fully dynamic and effective software engineering tool usage. This involves an AI system seamlessly and proactively integrating appropriate tools depending on the task at hand. There are a few challenges towards achieving this goal. First, the AI system must identify which tools could potentially be useful for the task at hand. Second, the system then needs to decide when to invoke the tool. A complex debugging task might require the use of `pdb` or `gdb` to track intermediate program states, while looking at input-output pairs may be sufficient for simple debugging tasks. Third, the agent then must figure out how to invoke the tool. If the agent knows that a certain function in a program has an error, it may wish to walk through only that function instead of the entire code from start to finish. Finally, the agent needs to incorporate the output provided by the tool in order to inform its next steps, e.g. edit the code if a bug was uncovered or run the tool again otherwise.

### 3.3. Interaction

While AI coding systems are increasingly more powerful, the majority of them are still at a low to medium autonomy level, serving as engineer assistance rather than achieving high or full automation. We identify a few key challenges of today's AI coding systems that prevent these systems from working with humans effectively at higher levels of autonomy.

**Vague Specifications and User Misalignment**: When using code LLMs or coding agents, we typically prompt them with a natural language specification. This can include a natural language description of the desired code, input-output examples, relevant code snippets, and other functional requirements. However, there is a gap in the level of abstraction between English and code, leading to incomplete or ambiguous specifications. This issue becomes more pronounced in longer programs, where the number of ambiguous decision points increases, and choices traditionally made by humans are instead implicitly embedded in the LLM's generated code. Consequently, users often experience misalignment due to vague specifications. While many code LLMs support multi-turn interactions, it remains inherently challenging for users to articulate their thought processes into follow-up natural language instructions.

*Inherent trade-offs in software development*: Designing large software systems always surfaces trade-offs between various desiderata such as readability, scalability, performance, maintainability, reliability, security, etc. These trade-

offs are often context-dependent. A long-term and rapidly moving project may be willing to trade off some performance to have simplicity and readability. Performance-critical applications may completely sacrifice readability to eke out every millisecond of speed (such as using bit-twiddling hacks). Finding the sweet spot among these trade-offs can often involve extensive prototyping and bench-marking to understand the performance characteristics of different approaches. However, user specifications in the initial prompt rarely include details about these trade-offs, nor do models often take them into account.

**Lack of Controllability**: When using AI coding systems, programmers often seek specific functionality, yet they lack reliable ways to steer LLMs toward generating precisely the desired code. Instead, they typically rely on a trial-and-error approach, repeatedly sampling outputs or providing feedback until the AI produces an acceptable solution. Consequently, significant human effort is still required to review and modify the code to ensure it meets the intended functionality (Weisz et al., 2024).

A way to improve controllability is for AI coding systems to recognize when human input is needed and communicate effectively—-yet this remains the top-reported challenge in human-agent collaboration (Shao et al., 2024a). LLMs rarely defer to humans for clarification, while developers often ask questions to clarify the description of a task provided by their peers. In addition, based on its knowledge of existing software, AI should incorporate implicit priors from a specification while keeping the user in the loop.

### 3.4. Measurement

**Task Diversity and Capability Isolation**: Current coding evaluations primarily focus on the code generation task, with little attention towards the tasks discussed in Section 2. As more agent-based approaches are introduced for software engineering (e.g. pairing a code generation model with a debugging model), these engineering-related capabilities beyond just code generation will be important towards designing a maximally performant system. Solely relying on end-to-end coding evaluations that focus on the overall correctness of a codebase makes it difficult to precisely measure progress and learn from the failure modes on individual tasks.

**Contamination**: Data contamination is a serious issue that, if not taken into account, can affect the soundness of various conclusions drawn from a set of benchmark results. In coding, the performance of LLMs on competitive programming (Xu et al., 2024a; Jain et al., 2024b) and SWE-Bench (Aleithan et al., 2024) tasks has been shown to degrade over time, indicating the possibility of older problems being contaminated due to public exposure on the internet. For simpler function-level HumanEval style problems, Matton

et al. (2024) suggest three potential causes of contamination: direct data leakage (benchmarks are on GitHub), synthetic data leakage (there are only a limited number of interview problems), and overfitting to test sets (benchmark hacking). In addition, for code, contamination can be hard to detect, as semantically equivalent code that is syntactically distinct could be thought of as contamination (Riddell et al., 2024). A recent benchmark, the Konwinski Prize[1], is a promising way to fairly evaluate SoTA LLM models by only using new GitHub issues.

**Construct Validity**: Construct validity refers to how closely a measurement reflects the underlying concept. Given the implications of rapid performance improvement in the AI for the code domain, it is essential to have high-construct validity benchmarks evaluating how well programming agents can perform. While benchmarks like SWE-Bench come close, user experiences do not currently match rapid performance gains obtained from them. This is partially because many desiderata in software engineering cannot be described cleanly via automated unit testing. Things like multi-turn code generation, designing an UI, and writing clean and idiomatic code are all difficult to quantitatively measure with precision. Designing reliable proxy metrics for these desired goals remains a challenge.

## 4. Paths Forward

Now, we describe several paths forward to address the aforementioned challenges. We group these into three parts: data collection, training, and inference time approaches.

### 4.1. Data Collection

When developing LLMs for code, the open-source community relies on datasets like *the Stack* (Lozhkov et al., 2024), consisting of trillions of GitHub tokens. However, richer sources of data exist such as fine-grained data of the developmental process. As this data needs to be collected on a large scale, we call for a community-based open source effort to assemble an open dataset for training code LLMs.

**Automatic Data Curation**: The advantage of code is it is possible to achieve strong, verifiable feedback with test cases, program execution engines, and other symbolic tools. Modern programming tools allow us to extract rich semantic and structural information about code.

*Augmenting Data with Program Information*: One way to increase semantic richness of coding datasets is to augment training datasets with detailed annotations describing various properties of programs. We hypothesize that this augmentation will significantly improve a model's understanding of code, leading to better generalization and stronger

---

[1] https://www.kprize.ai/

coding capabilities. This information includes static analysis (abstract syntax trees), program instrumentation, dynamic analysis, and formal verification.

*High-quality, Verifiable Synthetic Data*: The verifiability of code makes it possible to generate a large batch of data and filter out low-quality samples, even at the repository level. With the supervision of experts in various domains, we believe the community can come up with high-quality synthetic data pipelines with broad coverage of software engineering tasks. For example, to generate code with interesting program invariants, we can sample a large batch of programs, run an invariant detection engine, and retain only programs with interesting invariants.

**Human-Centric Data Curation**: While automatic data curation can provide a lot of additional information, there are still signals that are best obtained via manual efforts. We list three such categories of tasks below:

*Fine-Grained Data of the Developmental Process*: IDE providers have an advantage when it comes to coding data, as they have direct access to the full history and evolution of a codebase. Companies such as Google and Meta (Chandra, 2024; Murali et al., 2024) collect fine-grained signals such as edit history. With direct access to the full history and evolution of a codebase, they can track which suggestions are adopted over time. However, companies generally collect data for their own use cases, making an open-source effort invaluable for curating a collective and open use dataset.

*Data for Diverse SWE Tasks*: Most of today's code LLM training recipes focus on code generation because large-scale datasets are mostly in a continuous format. We believe that involving domain experts and curating data for all sorts of tasks (including challenging and out-of-distribution ones) will lead to diversity and more general coding capabilities.

*Human-Centric Data*: Code LLMs are trained and evaluated on carefully curated datasets with clear instructions and verifiable test cases. However, these models are often deployed in real-world scenarios where users provide vague specifications or incomplete requirements in their queries. Collecting human-centric data reflecting real-world model usage is a promising approach to bridging the gap between model development and deployment, but we currently have very little data of this form. An open-source community effort can lead to tools and environments such as open-source IDEs to capture real-world interactions in diverse modalities.

### 4.2. Training

**Environment Design for Code RL**: In recent months, RLVR has seen success in solving algorithmic programming problems through DeepSeek-R1 (DeepSeek-AI et al., 2025) and OpenAI o1. Recently, on SWE-Bench, SWE-RL (Wei et al., 2025) use RL on a rule-based reward to im-

prove performance on SWE-Bench. We find it promising to continue scaling the RL approach and create environments from tasks collected from real-world software engineering repositories. These environments can be used to improve reasoning skills, environment-interaction capabilities and tool usage. However, scaling this up significantly requires solving several research and engineering problems. First, installing arbitrary repositories from Github, even using CI is challenging and we require smarter solutions potentially involving LLM-based installation agents. Next, setting up execution infrastructure would require storing installed repository images in something akin to `docker` for efficient storage and fast container startup times (Team et al., 2025). Notably, combined docker images can grow massively large and often grow at hundreds of gigabytes even at a modest scale of a few hundred repositories. Because of the scale and engineering effort required, we advocate for the community to come together and create a unified gym environment for coding.

**Adapting to Specialized and Quickly Changing Codebases**: Being able to quickly adapt to code is an important and crucial skill, as codebases are often out-of-distribution and frequently change. Here, we describe three potential research directions leading to such adaptation.

*Test-time training (TTT) to custom codebases*: TTT is the recent paradigm of adapting to a specific problem instance by training on a narrow set of in-distribution examples (Akyürek et al., 2024; Sun et al., 2020). This can be used when working in a low-resource context, for example training on a specific codebase, new domain, or unseen API. One way to get data in these contexts is to collect and learn from trajectories of a SWE agent's behavior. We can keep track of previous model attempts and failures to prevent repeated mistakes. This will steer the model closer to the desired distribution, such as generating code in the specified version of libraries in the current context.

*Keeping an information bank of code information*: For library and versioning issues, retrieval can be very effective for preventing hallucinations of wrong versions of libraries, which can inherently lead to better synthetic data and agentic trajectories. During the TTT process, we can also keep a large growing memory bank of code, documentation, synthetic code, and agentic trajectories in the specialized context. Retrieving from the memory bank would improve the success of generating code, which can then be augmented to the memory bank, and so on, continuously increasing the amount of data and knowledge.

*Learning on the fly*: When humans are faced with a task they have never seen before, they are often able to draw from past experiences and quickly adapt and generalize to the new domain. This is one of the big unsolved challenges of today's LLMs: given an OOD coding task, how can models get up

to speed and productively work on the task with few samples? On toy domains, an example of this is DreamCoder (Ellis et al., 2021), a system that learns to solve problems by writing programs and automatically discovering domain concepts and composing concepts together. Designing such approaches for more practical applications is an exciting research direction that will have drastic implications for coding and reasoning.

**Training Code LLMs to Collaborate with Humans**: As we collaborate with AI more to write code, it is crucial that these models have capabilities that allow for effective collaboration.

*Specifications Beyond Natural Language*: While natural language prompts offer intuitive and flexible ways to express requirements, they are often ambiguous and incomplete. To address this, we can train models to use more precise and verifiable representations such as formal specifications and test-based specifications. Formal specifications allow an AI system to precisely verify correctness. Test-based specifications range from input-output examples and assertions to property-based tests. However, hand-crafted test suites can be incomplete, leading to misalignment where AI-generated code passes tests but does not genuinely meet functional requirements. Moving forward, we should train models to generate high-quality test cases based on the user's initial query, ensuring more comprehensive specification coverage.

*Learning to Quantify Uncertainty and Communicate Proactively*: As AI coding systems are increasingly deployed to complex SWE tasks, they encounter more ambiguous and uncertain scenarios compared to traditional benchmarks for coding models. Ideally, in such situations, these systems should proactively communicate with users to clarify tasks and acknowledge its own limitations rather than becoming stuck in endless failure loops or generating buggy code. A key challenge is enabling models to distinguish between well-specified and ambiguous instructions while quantifying uncertainty in a robust manner, which will require incorporating corresponding reasoning data into post-training. Another challenge in human-agent collaboration is communication (Shao et al., 2024b), highlighting the need to improve the proactive communication capability of the models. Current models often fail to ask meaningful questions when user input is ambiguous or insufficient and struggle to provide progress updates or verify plans in interactive settings. Enhancing the proactive communication skills of models requires innovative approaches to reward behaviors that yield benefits over multiple steps. Since communication with users does not immediately resolve the task at hand, but may improve long-term results, effective strategies must account for delayed rewards in training.

### 4.3. Inference Time Approaches

In this section, we advocate for several inference-time research thrusts that have currently been underexplored. Making progress on these fundamental directions can unblock many of the challenges previously mentioned.

**Semantic-Aware Embeddings and Retrieval**: We saw that code that is close in embedding space is more often syntactically similar than semantically similar (Zhao et al., 2023), making it hard to retrieve semantically similar code. However, before the LLM era, there were a variety of efforts to incorporate code properties when training embeddings. For example, Nye et al. (2020) train neural modules to represent program operations, leading to compositional program representations that encode the semantics of the underlying programming language. Many other works (Zohar & Wolf, 2018; Ellis et al., 2019; Chen et al., 2021b) attempt to learn execution-aware latent representations for partial and full programs, taking semantics into account.

Incorporating these techniques to train models to have better and more semantically aware representations can lead to models with a more general understanding of code (Sec. C.6). For example, if correct and buggy programs could hypothetically be separated in embedding space, then models could be steered away from the incorrect program space. While a complete separation might not be possible, having semantics-aware embeddings could lead to downstream improvements on SWE tasks.

*Retrieving via on-the-fly navigation*: Instead of keeping track of embeddings, another approach is to find retrievals on-the-fly by navigating the codebase. This prevents the high cost of continuously maintaining and updating the retrieval bank. An agent can learn to use command line functions such as `cd`, `ls`, and `grep`, and IDE functions such as jumping to function definitions or finding all references of a function. Static analysis tools can also be paired with the agent to improve code navigation, such as providing the abstract syntax tree (AST) or file structure of a codebase.

**Integration with SWE Development Frameworks**: As AI improves at SWE, there are increasingly more opportunities to incorporate it into the continuous integration and continuous deployment (CI/CD) process. In CI/CD, automated pipelines are the backbone for building, testing, and deploying code changes. CI/CD accelerates feedback cycles and minimizes integration issues. AI could contribute in many ways. First, AI-powered code review tools can be incorporated into CI pipelines to automatically identify and flag style violations, potential security vulnerabilities, and code smells before human reviewers are involved. Second, AI can provide intelligent deployment risk assessments. By analyzing code changes, test outcomes, and historical deployment data, AI can predict the likelihood of deploy-

ment issues, informing decisions about whether to proceed with automated deployment or mandate manual verification steps. Finally, AI can automate the generation of release notes by summarizing commit messages, issue tracker data, and relevant code modifications within the CI/CD process.

*Steering away from software anti-patterns*: In SWE, certain anti-patterns frequently lead to bugs. Common weakness enumeration (CWE) is a categorization of software and hardware weaknesses often leading to vulnerabilities. Because publicly available GitHub code often contains code with anti-patterns, bugs, and CWEs, LLMs often write code susceptible to these issues (Asare et al., 2023; Fu et al., 2023). Explicitly steering models against these vulnerabilities can lead to more secure and correct code, e.g. by collecting programs violating each CWE (synthetically or via GitHub) and using them as negative signal during SFT or RL.

**Neurosymbolic Approaches**: Code is a unique domain because there is a vast body of techniques from programming languages (PL) research to build off of, but the majority of AI for code research today does not leverage the symbolic properties of code. Traditional PL approaches have a few common shortcomings. First, they often require very complete and precise specifications. Many tools need to have specifications for all library functions, need to specialize to a precise version of the language, and need to specialize to the build system. Second, there is often a high computational cost due to the large search space. Third, there can be many false positives due to the limitations of the tool. We believe that deeply integrating these symbolic tools with LLMs can partially mitigate these challenges.

We provide a few examples of this potential integration. When generating code, program analysis techniques could be applied on shorter snippets of AI-generated code to surface potential bugs or prove properties. To improve general code understanding, LLMs can be trained with information about program structure such as ASTs (Gong et al., 2024). When debugging a large codebase, AI could be first used to narrow down potentially problematic sections of the code which are then handed off to PL tools for debugging. During code generation in DSLs, LLMs can leverage the grammar of the programming language to do constrained decoding (Poesia et al., 2022; Geng et al., 2023; Wei et al., 2023b) to mitigate syntactic errors.

**Scaffolding Human Supervision**: Once code LLMs are deployed for inference, it is crucial to scaffold human supervision of AI-generated code. This goes beyond merely enhancing the accuracy of AI-generated code, as humans often still need to make the final decision on whether to accept the code or understand it for future integration and maintenance. A study on Github Copilot usage (Al Madi, 2023) revealed that programmers tend to allocate less visual attention to AI-generated code. While we could train hu-

mans to better identify issues in AI-generated code (Singhal & Kumar, 2023), a more desirable approach is to design AI systems that scaffold human supervision, reducing human cognitive load when reviewing generated code.

One way to achieve this is by enriching AI-generated content with additional contextual information. In software engineering specifically, Sun et al. (2024b) highlighted the benefits of high-quality source code summarization in aiding software developers in understanding and maintaining machine-generated code. Second, interactive approaches can also enhance supervision such as *Live Programming* (Ferdowsi et al., 2024), a continuous display of the runtime values of a program. Finally, improving the readability and interpretability of AI-generated code itself presents a promising direction. Expanding on these ideas, future research should prioritize human interpretability in the design and optimization of AI coding systems, fostering greater trust and control in AI-assisted software development.

## 5. Alternative Views

An alternative view, particularly within the industry, suggests that limited further innovation is needed to achieve AI software engineers. They believe that scaling up larger models and more data is sufficient for creating AI engineers capable of contributing effectively. This is supported by the significant progress in recent years, for example the SWE-bench score increasing from an initial 2% to now 73%. However, we believe this view is too optimistic. As we have outlined in the paper, there are many fundamental capabilities in AI for SWE that today's models lack. As we argued in Sec. 4, improving these aspects requires fundamentally new forms of data, training environments, and inference-time approaches. The alternative perspective, while not unreasonable given recent progress, should not overshadow the need for more efforts in what we outline. As a research field, AI for code still has plenty to offer and overcome.

## 6. Conclusion

In this position paper, we identify key tasks at the heart of AI for software engineering and highlight critical cross-cutting challenges that permeate throughout many tasks. To drive progress in the field, we also pinpoint a set of exciting and promising research directions for alleviating these challenges and advancing AI towards being a more capable software engineer. We hope this work provides valuable insights about the current landscape of AI for SWE and encourages future research in these directions. By building on these insights, we are optimistic that we as a community can work together toward developing AI-driven solutions that better support software engineers in real-world settings.

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

## A. Introduction

AI for software engineering has made remarkable progress recently, becoming a notable success within generative AI. Despite this, there are still many challenges that need to be addressed before automated software engineering reaches its full potential. With additional efforts, it should be possible to reach high levels of automation where humans can focus on the critical decisions of what to build and how to balance difficult tradeoffs while most routine development effort is automated away. Reaching this level of automation, however, will require substantial research and engineering efforts across academia and industry. This paper provides an opinionated view of the tasks, challenges, and promising directions towards achieving this goal.

Many existing surveys overlap with the topics that are discussed in this paper. Liang et al. (2024) and Sergeyuk et al. (2025) survey the successes and challenges of AI programming assistants, (Wang et al., 2024c) survey using LLMs for software testing, and Joel et al. (2024) survey using LLMs in low-resource and domain-specific languages, and Zhang et al. (2023c) focus on automated program repair, both with and without LLMs. Finally, Yang et al. (2024d) is a roadmap for formal mathematical reasoning and has some overlap with our discussion on software verification.

In addition, many papers discuss the current state, challenges, and future of AI for software engineering (Fan et al., 2023; Ozkaya, 2023; Wong et al., 2023; Zheng et al., 2023; Hou et al., 2024; Jin et al., 2024; Wan et al., 2024b; Roychoudhury et al., 2025a). Our work draws inspiration from them, and we recommend that the reader consult with them for alternative perspectives.

In this paper, our goal is threefold. In Sec. B, we provide a structured taxonomy of concrete tasks in AI for software engineering. In particular, we emphasize that there are many other tasks in software engineering beyond code generation and code completion, encouraging research in these areas. We provide three measures for categorizing concrete realizations of each task: the scale of the problem, the logical complexity, and the level of human intervention.

Moving forward to Sec. C, we highlight nine challenges in the field that today's models face, each cross-cutting and applicable to several tasks. In Sec. D, we posit a set of promising research directions to tackle the challenges above, with Fig. 2 summarizing which directions correspond to each challenge. We hope that through our paper, the reader can appreciate the progress the field has made, understand the shortcomings of today's state-of-the-art models, and take inspiration from our suggested future ideas for tackling these challenges.

## B. Tasks in AI Software Engineering

We first provide a taxonomy of tasks in AI software engineering. To provide a structured way to consider concrete realizations of each task, we define three measures that apply across them: scope, logical complexity, and level of human intervention. To achieve an AI software engineer, we strive for AI to be capable across the board for all three measures.

**Scope Measure**: We define three levels of scope, the extent of changes that the AI makes to the codebase. *Function-level* scope refers to single, self-contained functions such as in HumanEval (Chen et al., 2021a) and MBPP (Austin et al., 2021). *Self-contained unit* scope goes beyond singular functions and to larger chunks of code such as entire files and classes, such as FullStackBench (Liu et al., 2024d) and BigCodeBench (Zhuo et al., 2024). Finally, *project-level* scope refers to larger codebases such as entire repositories, such as in Commit0 (Zhao et al., 2024) and SWE-Bench (Jimenez et al., 2024).

**Logical Complexity Measure**: Tasks require a wide range of reasoning abilities when it comes to devising algorithms to solve them. *Low logical complexity* tasks require little to no reasoning, such as writing CRUD (create, read, update, delete) applications or using APIs. *Medium logical complexity* tasks include most LeetCode problems, finding inputs to fuzz simple programs, and reasoning about execution behavior of multithreaded programs. *High logical complexity* tasks require meticulous and challenging levels of algorithmic and logical reasoning, either because the algorithm is complex or because the problem requires clever thinking and insights. This includes difficult competition programming problems, writing large thread-safe concurrent programs, cracking cryptographic ciphers, and solving SMT-like problems. Many popular coding benchmarks are function-level, medium-high logical complexity, such as APPS (Hendrycks et al., 2021), CodeContests (Li et al., 2022a), and LiveCodeBench (Jain et al., 2024b).

**Level of Human Intervention Measure**: AI coding is a collaborative task. Treude & Gerosa (2025) categorize interactions between developers and AI. Each interaction progresses through four phases: the *trigger* for the interaction, the *AI response* describing the system's output, the *developer response* capturing how developers react to the AI response, and the *output* of the interaction, the exact result. They characterize these developer-AI interactions into eleven types, including autocomplete

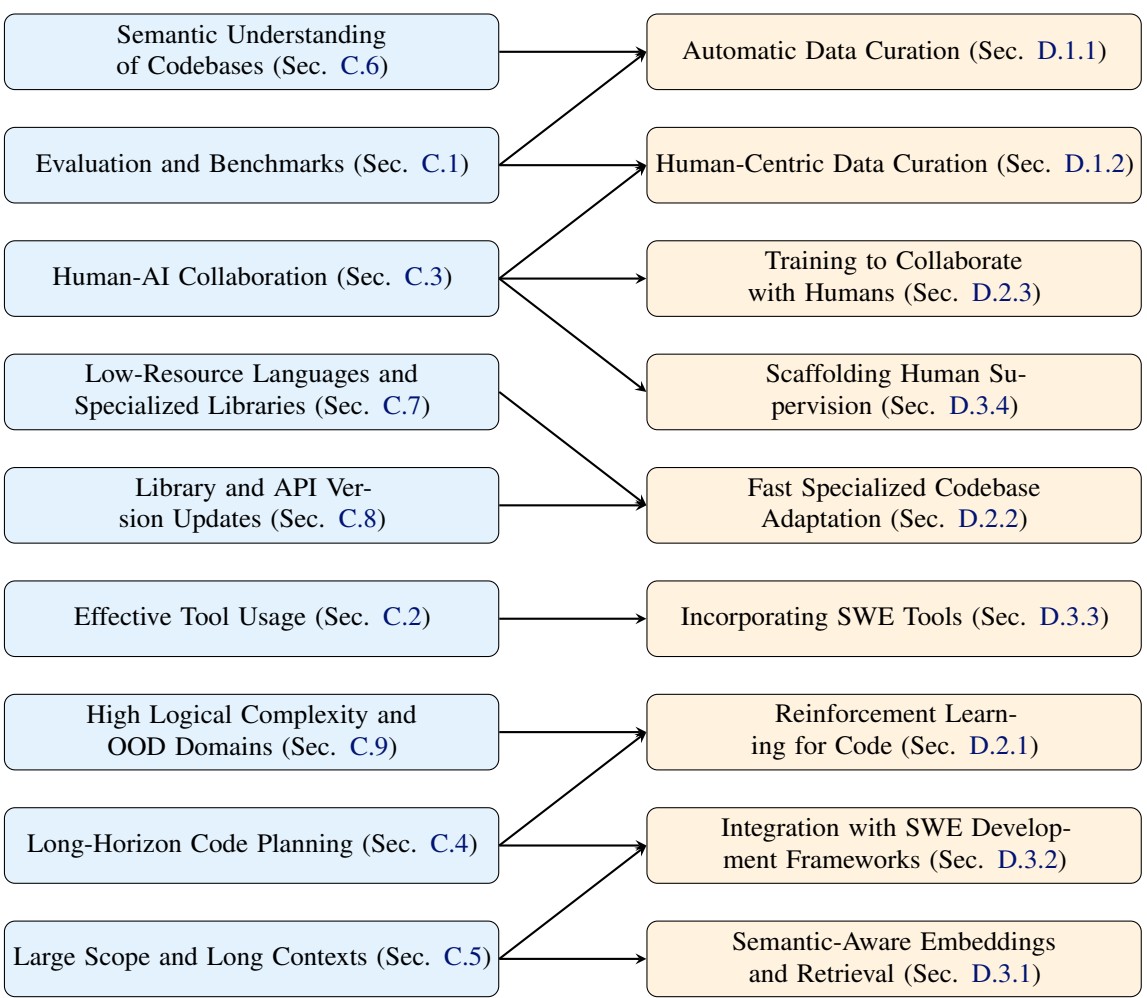

*Figure 1.* Overview of Challenges (Sec. C) and Paths Forward (Sec. D) in AI for Software Engineering

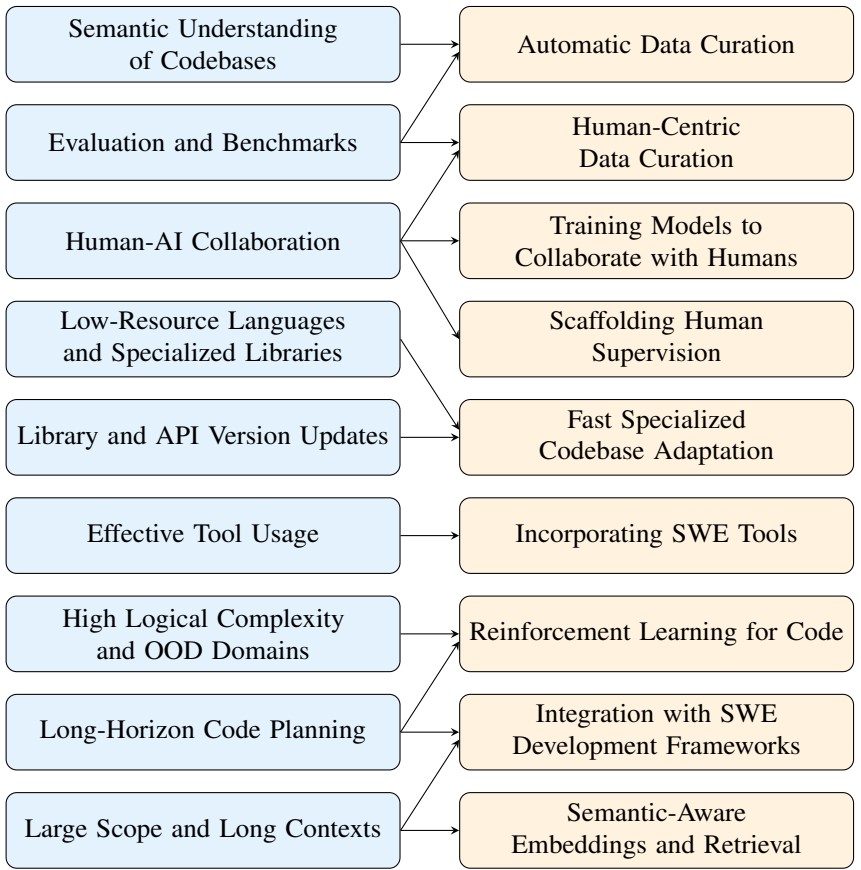

*Figure 2.* Overview of Challenges (Sec. C) and Paths Forward (Sec. D) in AI for Software Engineering

code suggestions, conversational assistance (e.g., asking a question about a codebase), selection-based enhancements (e.g., refactoring a selected chunk of code), comment-guided prompts (e.g., natural language to code), check correctness, and more.

We map these interactions to the autonomy taxonomy outlined in Morris et al. (2023)[2] to define three levels of human intervention. We distill their six levels of autonomy into three levels: low (*No AI* and *AI as a Tool*), medium (*AI as a Consultant* and *AI as a Collaborator*), and high (*AI as an Expert* and *AI as an Agent*). *Low autonomy* is when the human has full control over the task and uses AI to automate simple sub-tasks. This might look like writing a codebase with tests while leaving small function-level snippets for the AI to fill in. *Medium autonomy* is when the human and AI contribute a similar amount, with interactive coordination of goals and tasks. Here, the human and AI might both suggest refactorings and optimizations during the development cycle. *High autonomy* is when AI drives the interaction and tasks, identifying required changes and the changing demands of the user. The AI would defer to the human only when needed or for a final check, write the code and tests autonomously.

Next, with our taxonomy of measures in place, we turn to the set of tasks that are reflective of the tasks and capabilities of a human software engineer. We give a brief description of each task in this section, deferring a more extensive survey to Appendix H.

### B.1. Code Generation

Code generation is the task of generating code from a specification. In *code completion*, the specification takes the form of a preexisting code snippet, and the goal is to complete the snippet. The most popular form of code completion is tab completion, where the user can press the tab key to complete a block of code (e.g. GitHub Copilot). Tab completion is often done at line-level or function-level scopes but needs to be fast to provide users with a seamless experience. Another paradigm is *natural language to code*, where the specification is a natural language description with requirements such as the task description, input-output examples, or libraries to use.

Recently, AI-driven IDEs, such as Cursor Composer and Codeium's Windsurf Editor, have blurred the lines between the two paradigms. With the ultimate goal of decreasing the burden of human programmers, they aim to automatically infer the user's intent from the code context and user behavior (e.g. keystrokes, user edits, file navigation patterns). However, when intent is vague, they allow users to specify desired functionality via chat interfaces. Depending on scope and logical complexity, code generation can vary highly in difficulty. Reliable code generation in large codebases is still a challenge for state-of-the-art AI systems today.

### B.2. Code Transformation

#### B.2.1. CODE REFACTORING

In code refactoring, the goal is to take a working implementation of a piece of software and rewrite parts of it while maintaining correctness. One challenge with this task is that success extends beyond functional correctness or metrics. The goal is often to improve code maintainability, readability, or extensibility—qualities that can be inherently difficult to quantify and highly context-dependent.

For instance, extracting shared functionality into helper methods presents trade-offs between modularity and cognitive complexity (Parnas, 1972). While there are no hard rules for when to extract functionality, one heuristic adopted by software engineers is the rule of three ("three strikes and you refactor")–abstractions should only be used when a piece of code has been duplicated thrice. Because it can often be unclear what level of abstraction refactorings should be done at, completing a refactoring at a high autonomy level is also difficult. These challenges are further compounded by the need to understand implicit trade-offs customized to specific codebases, respect conventions, and reason about the long-term maintenance implications of structural changes. While code refactoring often has a low logical complexity, it can be laborious in practice due to scope, as seemingly small refactors can propagate across the entire codebase.

---

[2]We follow page 9, Table 2 from their paper

> *Example: React Fiber architecture refactor*: React's major refactoring was motivated by performance limitations in the original engine, particularly for complex UIs with animations and dynamic updates. Beyond challenges related to optimized implementation, a major challenge was providing backward compatibility while completely rewriting React's core algorithm. Being an open source tool, this refactor also required educating developers about new concepts without disrupting existing mental models highlighting nuances in real-world software system design.

### B.2.2. CODE MIGRATION AND TRANSLATION

An incredibly resource-intensive (time and manual effort) task frequently affecting companies is migrating large amounts of code while preserving all the original functionality and semantics. Such high-value migrations present opportunities for AI-assisted automation to reduce cost and manual effort. Code migration often has a very high scope (many files and systems affected alongside their interdependencies) and high logical complexity (semantic depth of required transformations, constructs in different languages may be different). Current solutions may excel at migrations with high scope but modest logical demands (API migrations, type conversions) but struggle with changes across component boundaries (Nikolov et al., 2025).

A special case of code migration is code translation (transpilation): rewriting code from a source language to a target language. In industry, this task can be motivated by several reasons, such as security and scalability concerns in legacy languages, avoiding the technical debt a project has accumulated over the years, and improving the performance of a codebase. Due to the safety-critical and cross-system nature of many migrations, this task often requires substantial human oversight in practice and cannot be done fully autonomously.

> *Example: Scala version migration*: A recent Scala 2.13 to 3 migration (Ricadat, 2025) illustrates these challenges, documenting a year-long effort. Critical issues included the loss of macro annotations, broken type projections, incompatible libraries, and compiler performance degradation—all requiring innovative workarounds and architectural changes. There have been many similar language migrations with analogous problems, famously Python 2 to 3 and Swift 4 to 5.

> *Example: COBOL*: COBOL powers 80% of in-person financial services transactions and 95% of ATM swipes while processing \$3 trillion in commerce a day, with over 220 billion lines of COBOL code in production (Taulli, 2020). However, there are less and less COBOL programmers, leading to the desire to migrate out of COBOL and into a modern language like Java (Sneed, 2001; Sellink et al., 2002; Sneed, 2010). However, because of the large scope and high logical complexity of existing COBOL code, migrating from COBOL to Java would be a monumental undertaking and many companies opt to continue using COBOL. These companies are still forced to migrate to newer versions like COBOL V6, because eearly versions of COBOL were gradually withdrawn from service. This task still requires skilled COBOL engineers and high precision, as it can often be difficult to understand the business logic of legacy code and introducing bugs can have dangerous implications.

> *Example: Twitter migration to improve latency*: Twitter[a] built its initial platform using Ruby on Rails, facilitating rapid development. However, as the user base expanded, performance and scalability issues arose. They migrated key components to Java and Scala, achieving a 3X latency drop. This transition required re-architecting the system to adapt Ruby's dynamic features to the statically typed environments of Java and Scala, exemplifying the complexities of large-scale code translation.
>
> ⎯⎯⎯⎯⎯⎯⎯⎯
> [a]https://www.infoq.com/news/2012/11/twitter-ruby-to-java/

> *Example: C to Rust*: There has been a push to use translation as a proactive approach to eliminate memory safety vulnerabilities in C-based systems. This has even garnered attention from the US Department of Defense[a], which has long-lived systems that disproportionately depend on C, supporting programs to translate C codebases to Rust (TRACTOR). Recent efforts like Syzygy (Shetty et al., 2024), C2SaferRust (Nitin et al., 2025), and AlphaTrans (Ibrahimzada et al., 2024) have shown the potential for hybrid approaches combining LLMs with traditional program analysis techniques. However, some significant challenges remain, including ensuring correctness in large codebases while maintaining desirable attributes such as speed, reduced vulnerabilities, and idiomaticity.
>
> ―――――
> [a]`https://www.darpa.mil/news/2024/memory-safety-vulnerabilities`

### B.2.3. CODE OPTIMIZATION

Transforming programs to improve performance characteristics while maintaining functional correctness is a critical software task. Optimizing real-world systems poses significant challenges due to the large scope and high logical complexity of the task, as performance bottlenecks must be identified and new algorithms to mitigate them must be proposed. Code optimization often has a large search and solution space with competing objectives like speed, memory efficiency, and readability, for example when optimizing kernel code at the PTX level for GPU-based AI model optimization (Zhao et al., 2025; Ouyang et al., 2025). In many scenarios, high levels of autonomy may not be desirable, as tradeoffs can depend heavily on external factors such as hardware, and the best optimizations may ultimately affect readability.

> *Example: Google Chrome performance improvements*: For over two decades, changes to the Chrome web browser have been an exemplar of optimization affecting real-world code (Chromium, 2018). Their V8 engine achieved a 20x performance improvement through coordinated optimizations across multiple layers - from implementing concurrent garbage collection that reduced bloat by 100x to developing specialized compilers like TurboFan that improved performance by 5-10%, to enabling background parsing and compilation that reduced compile time by 20%. The demand for cross-layer and low-level code changes (e.g., writing a new JavaScript interpreter) and building tools to measure and test representative performance metrics are key challenges for achieving this sort of real-world impact with LLMs.

### B.3. Software Testing and Program Analysis

In the process of software development, there will inevitably be bugs. The difficulty of detecting these bugs can vary depending on their scope and logical complexity. For LLMs, minor typos or correctness bugs (small scope, low logical complexity) are easier to spot (Mosolygó et al., 2021) while complex concurrency bugs and security vulnerabilities (large scope, high logical complexity) can be tricky because they can be hidden deep in the call stack, contain subtle logic errors, or be hard to isolate due to the large scope (Trent & Li, 2025).

### B.3.1. SOFTWARE TESTING

Software testing is a practical approach to prevent bugs, both during development and production. There are several popular approaches to software testing, some short-term and others longer-term. *Unit testing* refers to using input-output style tests that exercise the functionality of a piece of code. *Property-based testing* is based on formal specifications and relies on specifying test cases that ensure that known properties of the code hold. *Mutation testing* modifies a program subtly and ensures that the test suite can detect errors in these mutations. *Fuzzing* refers to executing programs with unexpected inputs and monitoring for exceptions, usually over a more extended time period.

*Example: OSS-Fuzz on FreeType*: OSS-Fuzz (Chang et al., 2024), Google's automated fuzzing infrastructure, has proven its value by swiftly uncovering security flaws in critical software. For instance, when a recent source change was made to FreeType—a font rendering library deployed on over a billion devices—OSS-Fuzz detected a heap-buffer-overflow within hours:

```
ERROR: AddressSanitizer: heap-buffer-overflow on address 0x615000000ffa
READ of size 2 at 0x615000000ffa thread T0
SCARINESS: 24 (2-byte-read-heap-buffer-overflow-far-from-bounds)
    #0 0x885e06 in tt_face_vary_cvtsrc/truetype/ttgxvar.c:1556:31
```

The goal of software testing is to design tests that can surface bugs reliably. This is evaluated through metrics such as code coverage–how much of the source code is executed when the test suite is run. An alternate to code coverage is mutation score, where mutants are generated, and the score is defined as the percentage of mutants causing the suite to fail. While practical, software testing faces challenges such as the scalability limits of traditional tools and the difficulty of manually designing tests with good coverage. As LLMs continue to improve at coding, they present a promising avenue for automatically generating high-quality tests.

*Example. Fault-based test generation at Meta*: Meta's Automated Compliance Hardening (ACH) system (Foster et al., 2025) is a system that generates tests aiming to catch real-world bugs. ACH works in three steps: first, the engineer describes the bugs they are worried about. Second, ACH combines LLMs with mutation testing to generate code with those bugs. Finally, these mutants were used to develop unit tests capturing them. ACH was used to generate tests for Messenger and WhatsApp, where engineers accepted 73% of its tests.

### B.3.2. PROGRAM ANALYSIS

While testing catches bugs, the most challenging software issues are security vulnerabilities and zero-day exploits, from memory corruption to privilege escalation. This requires a deep program understanding, that testing/fuzzing often misses. For instance, a *zero-day* is a vulnerability unknown to the software developers that is found by an attacker, and there is no patch available from the vendor. In such cases, the only practical approach is offensive security research, manual source code audits, and root cause analysis of prior vulnerabilities to harden codebases.

*Example: Variant Analysis:* Project Zero's (Hawkes, 2019) investigations at Google reveal that many in-the-wild 0-day exploits aren't entirely new—they're often variants of vulnerabilities that had been patched before. In their analysis of recent 0-day reports, nearly half of the issues were closely related to earlier bugs (such as those affecting Windows win32k and iOS IOMobileFrameBuffer). This finding underscores the importance of performing rigorous root cause and variant analyses. Instead of just fixing a single exploit path, security teams must comprehensively address the underlying bug class, ensuring that alternate exploitation routes are closed off for good–making this task more challenging.

Another example of a valuable but challenging analysis is invariant detection. A *program invariant* is a property of a piece of code that is guaranteed to be true at a specified program point, no matter what the input is. A simple example is that after the line `int x = c * c;` is executed, `x` must be nonnegative. Identifying invariants in a program can be useful when testing, debugging, and modifying code. This task can be challenging because it requires reasoning abstractly about code execution across many different potential inputs and execution paths to determine what properties must hold for all possible inputs.

### B.3.3. PROGRAM REPAIR

*Bug localization* is a significant challenge in program repair, as pinpointing the exact site of a bug can be challenging, especially in large codebases. Issues like out-of-memory accesses often manifest themselves further downstream, making it difficult to identify the root cause. Once the bug is localized, the next step is to repair the bug. LLMs can be an ideal tool for this because they have seen a wide variety of bugs during training. Function-level, low-logical complexity bugs can often be easily fixed by feeding back error information to the model. It can be tricker to perform repair in larger scopes (e.g. repositories) where the code has higher logical complexity. This can often require several steps, including designing and

implementing new algorithms or making complex refactorings across multiple files.

## B.4. Software Maintenance

### B.4.1. CODE DOCUMENTATION AND SUMMARIZATION

To ensure maintainability, readability, and ease of collaboration, code must be well documented. Good documentation needs to be written cleanly and crisply, describing what the function does and how the function works. It must also anticipate and address any misunderstandings that a programmer might have, such as potential side effects or special cases. Humans often see documentation as a chore and neglect it, leading to code and documentation frequently being out of sync. This has led to the concept of "self-documenting code", code that clearly conveys its purpose. As documentation is generally a task that has a low logical complexity and does not require too much human intervention, LLMs can help ensure that documentation is a continuously updated artifact in sync with the code.

### B.4.2. PULL REQUEST (PR) REVIEW

Reviewing pull requests is an integral aspect of the software development cycle. While the most essential requirement for PRs is that a new feature is implemented correctly, other important considerations include checking whether the repository's style conventions are satisfied, ensuring that the PR does not introduce any new bugs, verifying that program invariants and guarantees still hold, and inspecting whether tests are robust. Generally, reviewing PRs is a task requiring low logical complexity and can be automated relatively easily.

### B.4.3. CODE UNDERSTANDING, NAVIGATION, AND QUESTION ANSWERING

When encountering a codebase for the first time, developers often find it challenging to understand and develop a good mental model of the code. This can be due to many reasons: too many wrapper functions, excessive error-handling boilerplate, deep call stacks, or poor code cleanliness. One important challenge in code understanding is *code navigation*: finding where relevant functionality is implemented. Doing this well requires understanding the high-level layout of where every functionality lies in the codebase and the low-level understanding of which helper functions are used to implement each functionality.

Another challenge is *code question answering*: answering complex questions about a codebase, which requires sophisticated code understanding and reasoning abilities. Models should not hallucinate or give incorrect information that skews a developer's mental model of the code. Beyond other tasks mentioned in this section, developers might commonly ask questions related to data flow (when and where data structures get mutated), code functionality (whether there are any side effects), performance characteristics (determining the runtime and memory complexity of a function), or error handling (whether certain corner cases are handled).

## B.5. Scaffolding and Meta-Code

For a software system to work, the core logic must be written well, but that is not enough. Many infrastructural aspects must be in place to support the software. We group these into two main categories: *scaffolding* and *meta-code*. We define *scaffolding* as a task outside of the code that must be done to get the software running properly. Examples of scaffolding include setting up Google authentication, subscribing to APIs, managing file storage, and generating API tokens. In contrast, we define *meta-code* to be code that is important to make the system work but does not actually participate in the execution of its main logic. Examples of meta-code include test harnesses, CI/CD code, Makefiles, Dockerfiles, sandbox databases, and preprocessors. Scaffolding and meta-code often are small in scope and have low logical complexity but can require a lot of domain-specific knowledge about the application, requiring human intervention.

> *Example. Configuration validation:* Ciri (Lian et al., 2024) is a tool that uses LLMs for configuration validation on open-source projects including Django, PostgreSQL, and Redis. They find that while Ciri excels at detecting misconfigurations of syntax and range violations, it struggles to detect dependency and version violations and is limited to a narrow range of misconfigurations. They also find that LLMs are biased towards more popular configuration parameters, which may lead to hallucinations in out-of-domain scenarios.

*Infrastructure-as-code and Security.* A particularly challenging case is generating *Infrastructure-as-code* such as

`Terraform`, where you specify the type of infrastructure specifications (such as AWS EC2 instances, Kubernetes clusters, S3 buckets, VPC buckets) as code and execute it to create the infrastructure. When generating such code, LLMs struggle with security configurations due to the complex interplay between service-level permissions (e.g., AWS resource access), resource-level permissions (e.g., specific allowed actions), and provider-specific security primitives like IAM roles, security groups, and network access controls.

> *Example. Distinguishing permission levels in cluster setup*: (Terrateam, 2024) show that on a task of bringing up a cluster, models fail to distinguish between ECS (Amazon Elastic Container Service) Task Execution Roles (for container operations) and Task Roles (for application-level permissions). This resulted in overly permissive policies where a single role was granted both image pull permissions and DynamoDB table access, violating principles of least privilege.

## B.6. Formal Verification

The task of formal verification involves generating checkable, mechanized proofs that can guarantee that a piece of code works as intended. There are two major classes of formal verification: full functional verification (FFV) and property verification (PV). In FFV, the goal is to design a complete and precise formal specification that captures the desired behavior of the implementation, such as fully verified data structures (mutable lists, trees, graphs, hash tables) (Zee et al., 2008). The main challenge in full functional verification is in correctly writing the specification so that all desired properties are specified. FFV generally has a high scope and medium logical complexity, as the properties to verify are often straightforward to write once the correct abstractions are found.

While FFV provides a complete set of guarantees, it is usually sufficient to opt for PV, where a few key properties of a system are proven correct. Examples include: ensuring that two threads do not simultaneously enter a critical section of a program, verifying that a complex program will always terminate, proving the absence of security vulnerabilities like buffer overflows, and guaranteeing memory safety. One challenge that makes PV difficult to use in practice is the issue of false positives, where functionally correct code often does not pass property checks. A prime example is Rust: while the powerful type system enforces many desired guarantees, code with correct semantics often does not pass type checks. Another challenge is that many standalone tools for PV are often semantics-dependent, which can make them hard to maintain as language semantics change.

> *Example. Costly disasters*: Formal verification of software is important in mission-critical applications such as aircraft software, as software bugs may lead to costly disasters. In the maiden flight of the Ariane 5 rocket, a floating-point conversion error caused it to explode forty seconds after liftoff. Another case is with the computer-controlled radiation therapy machine Therac-25, where concurrency bugs led to six people being massively overdosed, leading to serious injury and deaths.

> *Example. Verified Compiler*: CompCert (Leroy et al., 2016) is a formally verified optimizing C compiler that supports a restricted subset of C including most of the ISO C 99 language. CompCert has been formally verified using the Coq proof assistant (The Coq Development Team, 2024), eliminating the potential for compiler bugs.

While formal verification tools have begun to see adoption in industry, they has not yet become mainstream because of these challenges. Code LLMs could greatly ease this burden and make it more feasible to verify code at larger scales, especially verifying properties requiring lower logical complexity.

> *Example. Property Verification: Coverity*: Coverity is a static analysis tool meant to find generic errors (memory corruption, data races) and system-specific violations (e.g. function-ordering constraints). In their report (Bessey et al., 2010), they highlight two issues mentioned earlier: churn and false positives. The first issue, churn, deals with ensuring that the tool produces the same result both when the code base is modified and across different versions of the tool, making upgrades "a constant headache". The second issue is that when the false positive rate is more than 30%, users ignore the tool and real bugs get lost among these false positives.

# C. Challenges

While the field of AI for code has made fruitful progress, cutting-edge AI still struggles with SWE tasks, especially at larger scopes and higher levels of logical complexity. Next, we discuss ten key challenges in AI for code. Each challenge spans multiple tasks, and progress on any can lead to improvements on many tasks at once.

## C.1. Evaluation and Benchmarks

> Today's code LLM evaluations focus on a narrow set of tasks, suffer from potential contamination, and do not reliably measure real-world software engineering abilities.
>
> *Potential solutions*: D.1

Our taxonomy of tasks and measures highlights some of the shortcomings of today's evaluations and benchmarks. For example, the majority of today's coding evaluations have no level of human intervention, with a few, such as Copilot-Arena (Chi et al., 2025), having low to medium autonomy. HumanEval, MBPP, APPS, CodeContests, and LiveCodeBench are all at function-level scope, with low to medium-high logical complexity. Commit0 (Zhao et al., 2024), SWE-Bench (Jimenez et al., 2024), TestGenEval (Jain et al., 2024a), RefactorBench (Gautam et al., 2024), SWE-Lancer (Miserendino et al., 2025) are at project-level scope with low to medium logical complexity.

**Task Diversity and Capability Isolation**: Current coding evaluations primarily focus on the code generation task, while most of the tasks discussed in Section B are either not studied such as Code QA or only studied in limited scopes like EvalPerf (Liu et al., 2024c), vulnerability detection (Mei et al., 2024), formal verification (Sun et al., 2024a). As more agent-based approaches are introduced for software engineering (e.g. pairing a code generation model with a debugging model), these engineering-related capabilities beyond just code generation will be important towards designing a maximally performant system. Solely relying on end-to-end coding evaluations that focus on the overall correctness of a codebase makes it difficult to precisely measure progress and learn from the failure modes on individual tasks.

**Contamination**: Data contamination is a serious issue that, if not taken into account, can affect the soundness of various conclusions drawn from a set of benchmark results. In coding, the performance of LLMs on competitive programming (Xu et al., 2024a; Jain et al., 2024b) and SWE-Bench (Aleithan et al., 2024) tasks has been shown to degrade over time, indicating the possibility of older problems being contaminated due to public exposure on the internet. For simpler function-level HumanEval style problems, Matton et al. (2024) suggest three potential causes of contamination: direct data leakage (benchmarks are on GitHub), synthetic data leakage (there are only a limited number of interview problems), and overfitting to test sets (benchmark hacking). In addition, for code, contamination can be hard to detect, as semantically equivalent code that is syntactically distinct could be thought of as contamination (Riddell et al., 2024). A recent benchmark, the Konwinski Prize[3], is a promising way to fairly evaluate SoTA LLM models by only using new GitHub issues.

**Construct Validity**: Construct validity refers to how closely a measurement reflects the underlying concept. Given the implications of rapid performance improvement in the AI for the code domain, it is essential to have high-construct validity benchmarks evaluating how well programming agents can perform. While benchmarks like SWE-Bench come close, user experiences do not currently match rapid performance gains obtained from them. This is partially because many desiderata in software engineering cannot be described cleanly via automated unit testing. Things like multi-turn code generation, designing an UI, and writing clean and idiomatic code are all difficult to quantitatively measure with precision. Designing reliable proxy metrics for these desired goals remains a challenge.

## C.2. Effective Tool Usage

> While software engineers use a wide suite of programming tools when programming, most of today's AI coding systems do not invoke tools. AI needs to be able to select which tool to use, decide how to use it, and interpret the outputs in order to continue making progress on the task.
>
> *Potential solutions*: D.3.3

---

[3] https://www.kprize.ai/

Software engineering has witnessed the development of various open and proprietary tooling support for programming, debugging, analysis, and code management over time. For example, program analysis tools provide static and dynamic assurances on code correctness. Print statements and debuggers are used for dynamically analyzing and debugging programs at a fine-grained level. Beyond programming, such tools are richly integrated into the entire software development lifecycle, e.g., code navigation or search, reviewing code, CI testing.

There have been efforts combining LLMs with tools such as calculators and search engines (Schick et al., 2023; Patil et al., 2023). However, effective integration of LLMs with software engineering tools is a more challenging problem. Several early works have incorporated such tool feedback in code generation in an automated fashion, for example, linter or execution feedback in (Olausson et al., 2023; Zhong et al., 2024b; Gehring et al., 2024). However, these works do not actively *interact* with tools. More recently, programming agents have started incorporating tool use within their workflows termed as Agent-Computer-Interface (Yang et al., 2024b). These tools range from aiding in general search (`grep`), providing code editor for making changes (Wang et al., 2024g; Anthropic, 2024), language server for static analysis (Liu et al., 2024e), dependency analyzer (Bairi et al., 2024), terminal access for bash commands including code execution (Yang et al., 2024b), debugger (BigSleep, 2024).

**Dynamic and Effective Tool Usage**: While many efforts combine LLMs and agents with tools, they do not achieve fully dynamic and effective software engineering tool usage. This involves an AI system seamlessly and proactively integrating appropriate tools depending on the task at hand. There are a few challenges towards achieving this goal. First, the AI system must identify which tools could potentially be useful for the task at hand. Second, the system then needs to decide when to invoke the tool. A complex debugging task might require the use of `pdb` or `gdb` to track intermediate program states, while looking at input-output pairs may be sufficient for simple debugging tasks. Third, the agent then must figure out how to invoke the tool. If the agent knows that a certain function in a program has an error, it may wish to walk through only that function instead of the entire code from start to finish. Finally, the agent needs to incorporate the output provided by the tool in order to inform its next steps, e.g. edit the code if a bug was uncovered or run the tool again otherwise.

> *Example: Performance Instrumentation*: A common way to instrument software systems is known as *compiler-inserted program instrumentation*. CSI (Schardl et al., 2017) is a tool that inserts instrumentation hooks to track objects such as memory loads/stores, function entry/exits, and basic blocks. CSI contains tools like code coverage tools, a memory-operations counter, a performance profiler, and a call-graph generator. To use the tool, the user must follow the API in order to write hooks so the correct aspects can be profiled. Tools like CSI are very valuable when trying to improve the performance of a piece of code, but are not trivial to use. In order for an LLM agent to use CSI effectively, it must first familiarize itself with the CSI API. Then, it needs to know exactly which aspects of the code to instrument, such as placing hooks before and after a function suspected to be a bottleneck. Finally, the agent needs to learn how to use the output of the tool to inform its approach to the task, such as deciding whether a block of code can be further optimized after seeing its performance profile.

### C.3. Human-AI Collaboration

> Human-AI collaboration is still far from seamless. First, specifications written by humans are often vague and leave out many details, leading LLMs to produce code misaligned with humans. There is also very little controllability with coding LLMs, and today's human-AI collaboration interfaces are still limited.
>
> *Potential solutions*: D.1.2, D.2.3, D.3.4

While AI coding systems are increasingly more powerful, the majority of them are still at a low to medium autonomy level, serving as engineer assistance rather than achieving high or full automation. We identify a few key challenges of today's AI coding systems that prevent these systems from working with humans effectively at higher levels of autonomy.

**Vague Specifications and User Misalignment**: When using code LLMs or coding agents, we typically prompt them with a natural language specification. This can include a natural language description of the desired code, input-output examples, relevant code snippets, and other functional requirements. However, there is a gap in the level of abstraction between English and code, leading to incomplete or ambiguous specifications. This issue becomes more pronounced in longer programs, where the number of ambiguous decision points increases, and choices traditionally made by humans are instead implicitly embedded in the LLM's generated code. Consequently, users often experience misalignment due to vague specifications.

While many code LLMs support multi-turn interactions, it remains inherently challenging for users to articulate their thought processes into follow-up natural language instructions.

*Specifications beyond text*: While today's specifications predominantly rely on text, there are many domains for which pure text is insufficient as a specification. In domains like robotics, virtual reality, embedded devices, and user interfaces, specifications often need to be multi-modal (e.g. showing the model a picture of an UI to create) and world-interfacing (e.g. providing simulation code describing a robot will interact with its environment).

*Inherent trade-offs in software development*: Designing large software systems always surfaces trade-offs between various desiderata such as readability, scalability, performance, maintainability, reliability, security, etc. These trade-offs are often context-dependent. A long-term and rapidly moving project may be willing to trade off some performance to have simplicity and readability. Performance-critical applications may completely sacrifice readability to eke out every millisecond of speed (such as using bit-twiddling hacks). Finding the sweet spot among these trade-offs can often involve extensive prototyping and benchmarking to understand the performance characteristics of different approaches. However, user specifications in the initial prompt rarely include details about these trade-offs, nor do models often take them into account.

*Implicit constraints*: Aside from functional/semantic correctness, there are also often implicit constraints in writing code. For example, many companies such as Jane Street and Google have style guides, and many GitHub repositories explicitly outline style elements that new code ought to follow. (Zou et al., 2019) find that GitHub pull requests that are more consistent with the style of the existing code get merged faster. Additionally, corporations may enforces codes of conduct or compliance requirements at the code level. Furthermore, codebases follow common programming patterns or system design patterns that are implicitly specified by the way the current code is written. However, when using code LLMs, these constraints are often inferred incorrectly (Wang et al., 2024h).

> *Example: Serializer-Deserializer pattern for objects:* Consider the issue astropy-#14181 from the `astropy` Python library. The issue requests support for a new input file format (reStructuredText) to load astronomical data into the codebase more flexibly. While the issue does not mention it explicitly, as per common practices, developers implement read (deserializer) and write (serializer) operations when implementing support for a new file format. This ensures data can flow bidirectionally between the file format and the application's internal data structures. However, models evaluated on this issue, as part of the SWEBench benchmark, only implemented the `read` method.

**Lack of Controllability**: When using AI coding systems, programmers often seek specific functionality, yet they lack reliable ways to steer LLMs toward generating precisely the desired code. Instead, they typically rely on a trial-and-error approach, repeatedly sampling outputs or providing feedback until the AI produces an acceptable solution. Consequently, significant human effort is still required to review and modify the code to ensure it meets the intended functionality (Weisz et al., 2024).

A way to improve controllability is for AI coding systems to recognize when human input is needed and communicate effectively—yet this remains the top-reported challenge in human-agent collaboration (Shao et al., 2024a). LLMs rarely defer to humans for clarification, while developers often ask questions to clarify the description of a task provided by their peers. For example, when a product manager refines a requirements document, developers who are unclear about the scope or specifications ask questions and leave comments, which the manager resolves iteratively to disambiguate requirements (Nahar et al., 2022). Based on its knowledge of existing software, AI should be able to incorporate implicit priors from a specification while keeping the user in the loop. For instance, when designing an academic website, certain expectations—such as including a list of publications and contact information—are implicit. However, whether to include a person's GPA would require explicit clarification.

**Restricted Human-AI Interface**: Existing interfaces for code LLMs primarily manifest as intelligence features embedded within integrated development environments (IDEs). Treude & Gerosa (2025) establishes a taxonomy of developer-AI tool interactions, emphasizing low-level support mechanisms such as auto-complete suggestions, selection-based enhancements, and conversational assistance within the codebase context. While this taxonomy comprehensively covers existing AI coding systems that function primarily as engineering assistants, its applicability becomes questionable as these systems advance toward higher levels of automation. For instance, the ubiquitous "Tab" interaction paradigm prevalent in intelligent IDEs may prove inadequate when AI systems transition from completing developer-scaffolded functions to authoring the majority of the codebase autonomously.

Current interfaces for coding agents, such as Devin, typically stream raw actions without adequate context or explanation. Given that these agents can execute numerous actions rapidly, developers face significant challenges in effectively monitoring the process, implementing timely interventions, or reasserting control when necessary. This lack of transparency can also undermine trust in AI-generated code (Wang et al., 2024e). While human-AI interface design has received extensive attention in autonomous vehicle research (Benderius et al., 2017; Tinga et al., 2022), similar consideration for AI coding systems remains notably absent.

### C.4. Long-Horizon Code Planning

> Large software engineering projects often require long-term planning about the design and structure of the code. LLMs struggle at two key aspects of this: designing good, lasting abstractions and respecting modularity and code quality principles.
>
> *Potential solutions*: D.2.1, D.3.2

When working on large projects, engineers and tech leads often make complex decisions about how to design and structure the code to best support the various functionalities that will eventually be needed. To build a long-lasting software system, an engineer must know the potential paths that the system's evolution might take. This requires domain expertise and experience in how different code structures require different forms of extension. We believe that today's language models are unable to perform this level of sophisticated planning.

**Designing Good Abstractions**: One instance of long-horizon code planning is choosing the right abstractions from the outset. An API designed with good abstractions will allow new features to be implemented seamlessly with minimal user overhead, while an API designed with poor abstractions may lead to excessive code duplication, refactoring, or even debugging. We discuss two examples of this, library learning and data representation.

*Library learning*: Designing APIs and libraries are designed with useful abstractions often leads to more code reuse and more intuitive interfaces. The challenge of library learning is to derive a library of useful abstractions from a corpus of programs by abstracting out common reusable features (Ellis et al., 2021; Stengel-Eskin et al., 2024; Bowers et al., 2023). While the traditional library learning literature has focused primarily on code reuse, a truly effective library must also prioritize ease of use and maintainability, as well as be robust and adaptable to future extensions.

*Data representation*: The choice between data structures leads to a variety of trade-offs when it comes to performance aspects such as memory usage and processing speed. For example, database engineers need to decide between various data models, storage formats, and indexing methods to balance performance.

> *Example: Database Design for Web Applications*: Database engineers strive to design their databases in a way that minimizes memory usage and maximizes query performance (speed). To achieve this goal, the databases community has spent considerable efforts optimizing both the high-level data representation and the underlying data structures (Kraska et al., 2018; Hawkins et al., 2011). Consider the task of designing a database schema for a restaurant owner to manage their business: keeping track of customers, managing a rewards program, maintaining the restaurant's inventory of ingredients, etc. One important design decision to make is deciding on a schema: while having a `reservation` and `customer` table is fairly straightforward, should we include a table for customer reviews or simply add rating and review fields in the `customer` table? Another important design decision is choosing which database indexes to include. While choosing the correct indexes can speed up queries significantly, indexes cost additional memory and must be kept updated. Making decisions like these requires knowledge of the application, context, and the effects of each option.

**Modularity and Code Quality**: LLMs are trained and optimized primarily for code correctness with insufficient focus on other aspects of code like quality and maintainability. This is further exacerbated with large scale reinforcement learning being performed using test cases which can lead to unintended consequences regarding code quality, as correct but poor quality code is still often given a high reward. Empirically, it has been observed that LLM written solutions are often more complex than human-written counterparts. For example, Jain et al. (2024c) identified that LLMs prefer to repeat existing code instead of making use of abstractions in the existing code. One aspect of code quality is modularity, ensuring that code does not get duplicated too often. Here, Berlot-Attwell et al. (2024) identified that library or tool reuse is non-trivial for

LLMs in coding and formal math domains.

### C.5. Large Scope and Long Contexts

> Coding is a domain where required context lengths can be very long, as codebases can consist of millions of lines of code, posing challenges to today's models. In addition, today's retrieval-based methods are still limited: models often retrieve incorrect information and can still struggle with leveraging retrievals effectively due to the difficulty of code reuse.
>
> *Potential solutions*: D.3.1, D.3.2

**Large Scopes**: At the repository level, the tasks in Sec. B become significantly more difficult and require many steps. In code generation, user alignment can be an issue because there are many decision points and tradeoffs that can compound. In code refactoring, modifications will touch multiple parts of the codebase, and it can be tricky to keep the repository consistent. In code debugging, functions can be large and bugs can be nested deeply within stacks of function calls. In code navigation, because there are so many functions interacting in various ways, it can be difficult to know where each piece of functionality is implemented and how the code is pieced together.

Another issue with large scopes is large context lengths. Software engineering often requires dealing with very large codebases–for example, Google has repositories with over a billion lines of code (Potvin & Levenberg, 2016). As this is far too large for modern-day LLMs, choosing the correct context to include when using LLMs is important.

> *Example: Debugging Cloud Applications*: Organizations often rely on monitoring and observability tools to track the performance of their applications. One such tool is Datadog, an observability service for cloud applications that can monitor infrastructure, detect security anomalies, and track database performance. For larger applications with more moving parts, these logs can consist of thousands of lines of JSON payloads. For humans, sifting through these logs is usually a matter of searching for certain keywords that they know will appear in the logs. However, LLMs often have a hard time interpreting large amounts of logs like these.

**Limits of Retrieval-Augmented Generation (RAG)**: Retrieval-based algorithms have been the predominant way to deal with long-context coding issues. First, the retriever retrieves relevant functions. Then, the generator leverages the retrieval to improve generation. While RAG has proven effective in many NLP tasks such as question answering (Gao et al., 2023; Lewis et al., 2020), the code domain provides new challenges for these methods.

*Retrieval*: In most NLP tasks, the retrieval step can be done relatively well because keywords that are in the query are often keywords that need to be retrieved. Unlike answering NL questions, writing code often requires drawing inspiration from code snippets that may be completely different syntactically. This can include programs with similar semantics, algorithms, or API calls, all of which potentially have very little in common when it comes to syntax. For example, the implementation of Dijkstra's algorithm in a GPS navigation application can guide the implementation of a shortest-path algorithm in a social media application. Because retrievers often rely on syntactic matching, these relevant programs can be hard to retrieve (Ma et al., 2024; Utpala et al., 2023).

When deciding what to retrieve, it is also necessary to have a sufficient awareness of other parts of the codebase so that you know which building blocks are necessary to construct the new function. This can make the retrieval task relatively tricky, as shown by failure modes on two benchmarks, CodeRAGBench (Wang et al., 2024i) and BRIGHT (Su et al., 2024).

> *Example: Failure Case of Finding Relevant Files When Resolving Issues:* BM25, despite its widespread use in code retrieval, demonstrates limitations in scenarios that involve large and complex codebases. For instance, in chartjs_Chart.js-7951 from SWE-bench Multimodal (Yang et al., 2024c), BM25 retrieval using the issue description returns suboptimal results. The top-3 retrieved files from `src/` are `src/scales/scale.radialLinear.js`, `src/scales/scale.linearbase.js`, and `src/helpers/helpers.canvas.js`. However, the critical modifications required to resolve the issue should occur in `src/elements/element.bar.js` and `src/controllers/controller.bar.js`. This retrieval failure impedes the effectiveness of coding agents, many of which are augmented with code retrieval systems. When agents focus their attention on irrelevant files, their ability to resolve the issue successfully becomes substantially compromised.

*Generation*: In NLP tasks, the generation step often is a straightforward application of the retrieved information. However, in code, writing a new function requires more than copy and paste. This is closely tied to the problem of code reuse: piecing together relevant snippets of code in a precise and productive way to fit the current context. Depending on what is retrieved, each piece of retrieved content provides different information. This can include information about the language's syntax, documentation about the API, clues about the algorithm to be written, or examples of similar functionality being written. Ding et al. (2023) find that even when the oracle context is retrieved, LLMs tend to misuse it, highlighting a lack of semantic understanding, which we discuss in the next section.

> *Example: Bad Generation Despite Identifying the Correct Context:* Ding et al. (2023) highlights a failure case where a code LLM fails to complete a Python test case correctly, even though it has the correct context. The function name from the context, `test_case_convert_camel_to_snake`, suggests that the function being completed is a test case for `convert_camel_to_snake`. With the given context, the model generates the function as `convert_camel_to_snake`, which however does not match the larger codebase as other pieces of code expect this function name to be `camel_to_snake`. While this issue can partly be attributed to incomplete retrieval of relevant information, it also presents a challenge for code LLMs, as they must recognize such inconsistencies—especially when the immediate context is correctly provided—thereby avoiding high-confidence errors.

### C.6. Semantic Understanding of Codebases

> Being able to effectively write code relies on having a strong semantic understanding (somewhat like a world model) of the entire codebase: structurally seeing how various parts of the code go together, knowing what is implemented where, understanding how the algorithms work, and keeping track of program invariants at certain program points. LLMs struggle with this global semantic understanding
>
> *Potential solutions*: D.1

A global and holistic semantic understanding of a codebase is important for performing almost all code-related tasks. For example, let's say an engineer is asked to improve the runtime performance of a query. To do so, they must first understand the codebase's structure well enough to know where all the pieces of the algorithm are implemented. Then, they need to understand the algorithm and implementation in detail. This includes both the high-level algorithm (including its time complexity) and the low-level implementation details to identify both algorithmic and implementation bottlenecks. Finally, after coming up with a solution, they must then return to their understanding of the global code structure so that they can integrate their new algorithm without introducing new bugs.

LLMs struggle at semantic understanding of codebases for several reasons. First, the way that code is pieced together can be relatively intricate, and understanding all these complex relationships can be difficult. Second, code can often have units with high logical complexity that contain custom algorithms that may never have appeared anywhere in the training data. Finally, because a disproportionately large number of LLM training tokens are spent on code generation rather than other coding tasks, models may lack a holistic awareness and world model of code.

One desiderata is that models can generalize knowledge across various coding tasks (Roychoudhury & Zeller, 2025). However, this may not be straightforward as just training on more tasks: Gu et al. (2024) found that coding models fine-tuned on additional natural language/code pairs saw significant improvements on code generation but did not transfer to improving

code understanding and execution reasoning. While there have been successful efforts to augment code LLM training with execution information to improve general coding capabilities (Ni et al., 2024; Ding et al., 2024c), imbuing code LLMs with a general and holistic understanding of code remains an important challenge today.

### C.7. Low-Resource Languages and Specialized Libraries

> Models struggle with low-resource languages and codebases with specialized libraries. Because of the limited exposure they have in these contexts, models fail in a variety of ways including generating syntactically incorrect code, misunderstanding the semantics of certain constructs, and using libraries improperly.
>
> *Potential solutions*: D.2.2

As we adapt code LLMs to individual codebases, generating correct code in out of distribution (OOD) scenarios becomes crucial. Much of software development in business contexts revolves around proprietary codebases, which is a distribution shift from the open-source code that dominates LLM training data (Ahmed et al., 2024a). These OOD scenarios include domain-specific languages (DSLs), custom internal libraries, low-resource APIs, and company-specific coding styles and conventions.

**Syntactic Failures**: Models have been shown to hallucinate constructs from higher resource languages when working in low-resource languages. Blinn et al. (2024) remark that *"contemporary LLMs fail to follow Hazel's syntax and semantics, often borrowing syntactic forms and library functions from [higher-resource languages like] OCaml and Elm"*.

> *Example. Syntax error in Triton*: In Listing C.7, we show an attempt from Gemma-3 27B to write a dot-product kernel in a low-resource GPU programming language called Triton (part of docstring omitted for brevity). Gemma uses indexing notation such as `a[index]`, which is not a valid Triton construct. Models like o1 and o3, however, do not make this mistake.
>
> ```
> @triton.jit
> def dot_product_indexed_kernel(a, b, indexes, out, N):
>     """
>     Computes the dot product of two vectors a and b using an index vector.
>     ...
>     """
>     block_size = 64  # Adjust block size for optimal performance
>     grid_size = (N + block_size - 1) // block_size
>
>     block_id = tl.program_id(0)
>     start = block_id * block_size
>     end = min(start + block_size, N)
>
>     accumulator = tl.zeros(1, dtype=tl.float32)
>
>     for i in range(start, end):
>         index = indexes[i]
>         accumulator += a[index] * b[index]
>
>     tl.store(out, 0, accumulator)
> ```

**Poor Semantic Understanding**: In low-resource languages, models have less exposure to the various language constructs. Therefore, they have a weaker semantic understanding of the language. Many studies reveal that code LLMs perform poorly when asked to write code in low-resource languages. Due to the lack of training data in these OOD domains, models may struggle to write common primitives or piece together functionality coherently. On HumanEval, Qwen 2.5 Coder Instruct (32B) (Hui et al., 2024) has an accuracy of 83% in Python but only 27% in D.[4]

**Library Usage Failures**: In OOD scenarios, LLMs lack awareness of the libraries and functions available for use. In new

---

[4]As reported by the BigCode Models Leaderboard on the MultiPL-E benchmark (Cassano et al., 2023)

codebases using custom libraries, many functions appear only a few times, providing limited training data for AI models to learn their usage. This scarcity can lead to overfitting, where models fail to recognize an effective use-case of these functions. Models also frequently hallucinate non-existent functions based on patterns that it infers.

> *Example.  Hallucinating theorems in Lean*:  In Listing 1, the LLM makes up a theorem named `infinitelyManyPrimes_arithmeticSequence`. While the model understands that it should use Dirichlet's theorem, the actual theorem is named `forall_exists_prime_gt_and_eq_mod`. This theorem occurs very few times in all of the currently available Lean code on GitHub, so it is likely that LLMs did not train on it many times.
>
> ```
> theorem infinitely_many_primes_of_form_6k_plus_1 :
>   ∀ n : ℕ, ∃ p : ℕ, p > n ∧ Prime p ∧ ∃ k : ℕ, p = 6 * k + 1 := by
>   intro n
>   -- Use Dirichlet's theorem for arithmetic progressions
>   have h := Nat.infinitelyManyPrimes_arithmeticSequence 6 1
> ```
>
> *Listing 1.* LLM-generated Lean code snippet that hallucinates a nonexistent function

## C.8. Library and API Version Updates

> Software engineering has a unique property that repositories and libraries are constantly changing. Code LLMs have trouble adapting to these rapid changes, often struggling to use the correct version of libraries and ignoring new paradigms and features.
>
> *Potential solutions*: D.2.2

Continual learning, the idea of training an AI system to take in new information continually, has been a long-standing challenge in AI and NLP (Wu et al., 2024; Wang et al., 2024d). In software engineering, codebases are continuously changing as new features are supported and awkward design patterns are reworked. While backwards compatibility is often prioritized in software design, it inevitably becomes broken as codebases evolve further. Therefore, programming libraries have version releases, each release supporting and deprecating features in the last version.

There have been a few works exposing this issue. For example, CodeUpdateArena (Liu et al., 2024g) and GitChameleon (Islah et al., 2024) are two benchmarks exploring the ability of LLMs to write version-specific code, examining this issue at the function and file level. They find that language models struggle to adapt to these changes even with this limited scope. In theorem proving (Lean), Kumarappan et al. (2024) try to mitigate this by developing a lifelong learning framework that continuously learns and uses new theorems. In real-world engineering, the challenge of library and API versioning generally spans across an entire repository, as everything must be kept consistent. To our knowledge, there are no techniques that successfully deal with this challenge at such a large scale. This problem is difficult for a few reasons, which we discuss below.

**Version Identification**: In order to successfully deal with version changes, a LLM must first identify which version of each library is being used in a codebase. This may often be quite difficult, because versioning information can be hidden deeply within a codebase. Sometimes, it can be found in comments or configuration files, but in the worst case, it must be inferred from the library calls being used. To make things worse, some code may be compatible across multiple versions, while other code will cause errors only in specific versions. Therefore, the model will often require a deep understanding of both the codebase and the nuances between different versions in order to infer the version at hand.

> *Example: Debugging Frontend Code:* Frontend framework usually has more frequent versions update, making it hard for code LLMs to work with. For example, when helping a user debug the "NextRouter was not mounted" issue, Claude 3.7 tries various solutions without recognizing that the core problem requires importing `useRouter` from `'next/navigation'` instead of `'next/router'`, a crucial distinction since the user's codebase leverages App Router in Next.js 13.

**Version Adaptation**: Many fast-changing libraries are not backward compatible as older features become deprecated. It can be difficult for LLMs to implicitly keep track of which constructs and patterns are associated with each version. Therefore, consistently using constructs from the right version can be difficult. As we will see in the examples below, LLMs often write code that mixes and matches API constructs from different versions of the same library.

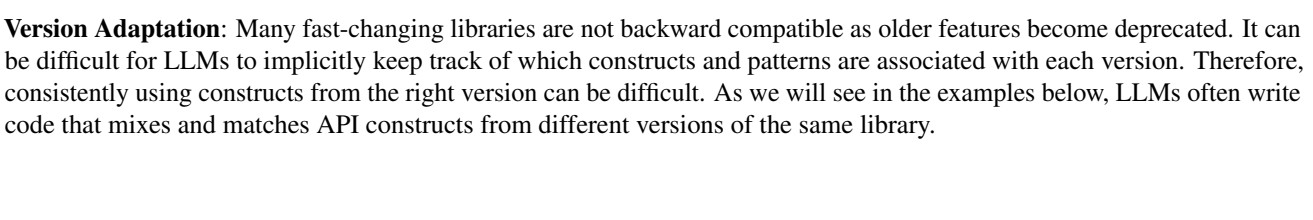

*Example. Typing Hints*: While Python 3.5 required importing types from the typing module, Python 3.9's PEP 585 enabled direct use of built-in types for generics (e.g., `list[int]` vs `typing.List[int]`). However, language models tend to default to the older `typing` module syntax.

**Continuous Adaptation to Paradigms, Features, and APIs**: New styles, patterns, and paradigms are often introduced to replace older, more cumbersome ways to write code. For example, React came out with its *Hooks* paradigm in version 16.8 (2019). Over the next few years, developers transitioned from the old class components paradigm to using hooks, as hooks made code cleaner and more maintainable. Only in early 2023, with the launch of `react.dev`, were Hooks the default paradigm in the documentation. For language models, incorporating these features can take a long time, because code in these new paradigms are initially completely absent in the training data and inherently in the low-resource regime. In Kharma et al. (2025), the authors find that LLMs fail to utilize security features in compiler and toolkit updates (such as in Java 17), still relying on legacy methods such as insecure random number generation. While it is possible to use retrieved examples and documentation in order to coerce language models to write code using new and updated features, we should strive to create AI coding assistants that can quickly internalize new changes and be able to naturally incorporate new features and paradigms, even without an abundance of training data. For each task, the language model should be able to reason about the best way to write the code, independently of the number of occurrences seen in the training data.

*Example. Lean 3 vs. Lean 4*: Lean (De Moura et al., 2015) is a programming language that allows users to write formal proofs of mathematical theorems. In 2017, using Lean 3, enthusiasts implemented a library for mathematics called `mathlib`, with over half a million lines of code. Because Lean 3 had many shortcomings, Lean 4 (Moura & Ullrich, 2021) was initiated at the beginning of 2021 to address many of these issues. There was a massive undertaking to port all of the `mathlib` code over to Lean 4, and only in September 2023 was there a stable release of Lean 4, the version of Lean that is predominantly used today. The two versions are generally incompatible. We hypothesize that, due to the recency of Lean 4, most language models have been trained on much more Lean 3 code compared to Lean 4 code. When asking models to generate code in Lean 4, it sometimes generates code with Lean 3 coding conventions. Other times, it uses theorems and lemmas from Lean 3 that are deprecated in Lean 4. In Listing C.8, we show an example of prompting `o3-mini` with a Lean 4 problem, where it generates Lean 3 syntax (e.g. `begin`).

```
Prompt: Write and prove the following in Lean 4: the sum of the first n positive
    ↪ integers is n*(n+1)/2. Begin with "import Mathlib". Surround your Lean 4 code
    ↪ with ```lean4 and ```
```

```
/-- The sum of the first n positive integers (i.e. 0 + 1 + ... + n) is n * (n + 1) /
    ↪ 2. -/
theorem sum_first_n (n : ℕ) : (range (n + 1)).sum id = n * (n + 1) / 2 :=
begin
  induction n with n ih,
  { -- Base case: when n = 0, we have range (0 + 1) = {0} and so the sum is 0, which
    ↪ matches 0 * (0 + 1) / 2.
    simp [range],
  },
  { -- Inductive step: assume the formula holds for n, and prove it for n + 1.
    -- First, note that range (n + 2) = insert (n + 1) (range (n + 1)).
    rw [range_succ],
    -- Since (n + 1) ∉ range (n + 1), we can express the sum as:
    rw [sum_insert (not_mem_range_self (n + 1))],
    -- Substitute the induction hypothesis:
    rw [ih],
    -- We now have to verify:
    -- n * (n + 1) / 2 + (n + 1) = (n + 1) * (n + 2) / 2.
    -- This equality follows by algebra (ring calculation).
    ring,
  }
end
```

## C.9. High Logical Complexity and OOD Domains

Tasks such as writing highly concurrent code or discovering performance optimizations have a high logical complexity, often proving difficult for even the best human coders. Similar to solving research-level math problems, these out-of-distribution domains are very hard for LLMs.

*Potential solutions*: D.2.1

Some programming tasks are challenging for even the best human programmers, requiring approaches with a very high logical complexity. Examples of tasks that fall into this category include superoptimizing programs, discovering attacks for purportedly secure code, writing performant compilers, optimizing GPU kernels (Ouyang et al., 2025), and writing very error-prone and very technical code.

*Example. Synthesis of Sorting Kernels*: An example of an out-of-distribution domain is synthesizing fast assembly code for sorting kernels. In 2023, AlphaDev (Mankowitz et al., 2023b) used reinforcement learning to find a SoTA kernel for sorting length 3-5 arrays. While this appeared to be a superhuman performance, shortly after, (Neri, 2023) hand-wrote a kernel shorter and faster than the one found by AlphaDev. Later, (Ullrich & Hack, 2025) developed an algorithm based on enumeration and intelligent heuristic-based sampling that beat both of these. In addition, the algorithm ran faster than AlphaDev by two orders of magnitude. In this case, while AI was able to achieve an impressive performance, humans were able to discover better algorithms.

*Example: Verifying File System Properties*: In formal verification, when working with new domains, it is necessary to devise new theories to faithfully represent desired properties. For example, FSCQ is a formally certified crash-proof file system with the provable guarantee that under any sequence of crashes followed by reboots, FSCQ will recover the file system correctly without losing data (Chen et al., 2015). In this domain, one challenge is that proving safety cannot be done at the source code level–because instructions are not atomic, data may be lost if the crash occurs within a non-atomic instruction. Instead, a new logic known as the Crash Hoare logic (CHL) needed to be developed, and constructs representing a crash condition and recovery procedure needed to be described. Constructing a logic like this would be very difficult for AI systems.

**Limits of Symbolic Techniques**: When it comes to applying symbolic techniques to these tasks, there are a few limiting factors that make them difficult to tackle. First, for synthesis-style tasks, the search space can be very large. Deductive and rewrite-based synthesis techniques are unable to explore a majority of the search space. Second, verifiers can be limited in power, such as when dealing with properties in concurrency or weak memory models. Third, many domains lack clean models to reason about properties, such as dealing with memory bandwidth in GPU kernels.

Because they are hard for humans, these tasks are very rarely in the training data of today's language models. They have unique, domain-specific, challenges that making generalizing from existing data difficult. For these problems, language models rely heavily on feedback-driven search algorithms (Mankowitz et al., 2023b), and it can be difficult to navigate the search space effectively. In addition, many of these tasks lack feedback mechanisms, which is crucial for AI to pick up learning signals. When designing a complex algorithm or data structure, it is often hard to know if you are on the right track until you get to the correct result. When writing code for a large multithreaded operation, it may be hard to know if the algorithm has concurrency issues until all the parts are fully fleshed out. Without feedback, incremental improvement is nearly impossible.

# D. Paths Forward

## D.1. Data Collection

One bottleneck in the development of AI for SWE in the open-source community is the lack of access to fine-grained and high-quality code data. In Sec. D.1.1, we discuss how automated techniques can mitigate this by augmenting existing programs with symbolic information and generating synthetic data with symbolic verifiers. However, there are other crucial signals in programming that can be hard to automate. We envision that a large community-based coding data curation effort will be very impactful. In Sec. D.1.2, we discuss examples of datasets that such a community could create that would unlock new capabilities in code LLMs.

*Challenges addressed: C.1, C.3, C.6*

### D.1.1. AUTOMATIC DATA CURATION

**Augmenting Data with Program Information**: One challenge in enabling LLMs to develop a world model of code is that programs are often treated like text: as tokens with no semantic information. However, modern programming tools allow us to extract rich semantic and structural information about code. By leveraging these tools, we can augment training datasets with detailed annotations describing various properties of programs. We hypothesize that this augmentation will significantly improve a model's understanding of code, leading to better generalization and stronger coding capabilities. Information can include:

- Static analysis: the syntactic structure of a program (abstract syntax trees, control flow graphs), information about the type of each variable, data flow analysis (reachability, liveness analysis)

- Program instrumentation: memory consumption, runtime analysis, aliasing, and code coverage (like statement or branch coverage)

- Dynamic analysis: program states at various points in the program, call stacks, dynamically curated properties (often relies on instrumentation)

- Formal verification: concurrency analysis, program invariants, loop invariants, memory safety

There have been a few examples of this in the literature: Ouyang et al. (2025) leverage profiler feedback to improve GPU kernel generation, Ding et al. (2024c;b); Ni et al. (2024) incorporate execution trace information, Pei et al. (2023) train with program invariants, GraphCodeBERT (Guo et al., 2020) incorporate data flow information, and Shypula et al. (2023) train on a dataset of performance-improving edits.

**High-quality, Verifiable Synthetic Data**: The advantage of code is it is possible to achieve strong, verifiable feedback with test cases, program execution engines, and other symbolic tools. This makes high-quality synthetic data generation viable, as it is possible to generate a large batch of data and filter out low-quality samples. For example, to generate code with interesting program invariants, we can sample a large batch of programs, run an invariant detection engine, and retain only programs with interesting invariants. While synthetic data in code has mostly been at the function-level scope, there are no fundamental bottlenecks to expanding to larger scopes. As code is quite compositional, individual building blocks can be combined to generate complex synthetic data at the repository-level scope, which can be very helpful in both training and evaluation.

While the importance of having high-quality data vs. high quantities of data is debated, using verified data has proven to be useful. For example, Liu & Zhang (2025) shows that simply removing bugs in existing datasets such as TACO (Li et al., 2023) can lead to significant boosts. KodCode (Xu et al., 2024a) also showed that fine-tuning on verified synthetic data also leads to significant improvements. However, these works work with programs at the function-level scope with low to medium logical complexity, and we imagine that general SWE abilities can improve with synthetic data across scopes and logical complexities.

In DSLs, where programs can be cleanly described with semantics and rewrite rules, one can symbolically generate programs with desired properties via sampling, drawing on enumeration techniques from program synthesis (Gulwani et al., 2017). This technique has been successfully applied to make considerable progress in difficult reasoning tasks such as ARC-AGI (Li et al., 2024d) and math olympiad problems (Trinh et al., 2024; Google, 2024; Chervonyi et al., 2025).

### D.1.2. HUMAN-CENTRIC DATA CURATION

Below, we list three classes of human-annotated data that would be invaluable for the next generation of coding LLMs.

**Fine-Grained Data of the Developmental Process**: Many code LMs are trained on datasets such as *the Stack* (Kocetkov et al., 2022; Lozhkov et al., 2024), consisting of trillions of tokens sourced from GitHub. However, training on raw GitHub tokens omits many crucial human signals in the process of software development. For example, companies such as Google rely on internally captured logs of high-quality SWE data. This includes "fine-grained code edits, build outcomes, edits to resolve build issues, code copy-paste actions, fixes of pasted code, code reviews, edits to fix reviewer issues, and change submissions to a repository" (Chandra, 2024). Similarly, Meta and GitHub Copilot use telemetry with their AI coding assistants to track and leverage signals from AI-generated code (Murali et al., 2024; Ziegler et al., 2024). These tools, along with coding IDEs like Cursor, could provide a treasure trove of reward data for RL-based methods. With direct access to the full history and evolution of a codebase, they can track which suggestions are adopted over time. However, collecting data from human usage also raises critical privacy and intellectual property concerns.

**Data for Diverse SWE Tasks**: Most of today's code LLM training recipes still focus primarily on code generation because large-scale datasets are mostly in a continuous, tokenized format. However, as described in Sec. B), there are many tasks involved in software engineering which models lack exposure to. Training on a broader set of tasks would also incentivize models to learn *general* software engineering capabilities beyond generation (e.g. a better understanding of program semantics). As initial evidence, (Li et al., 2025a) find that training models on input-output prediction data leads to consistent improvements on reasoning tasks.

The lack of high-quality data on these tasks makes it hard to train on them. It can also be hard to automatically curate them on GitHub. For example, for code refactoring (Sec. B.2.1), we need paired repositories before and after refactoring, ideally with the refactoring changes described. While some signal such as commit messages and version releases can be used, many repositories lack clean commit histories and releases conflate many features at once. Therefore, to mitigate this, we envision large community-based efforts curating task-specific data on these diverse challenges.

**Human-Centric Data**: Code LLMs are typically trained and evaluated on carefully curated datasets with clear instructions and verifiable test cases. However, as discussed in Sec. C.3, these models are often deployed in real-world scenarios where users provide vague specifications or incomplete requirements in their queries. Collecting human-centric data that reflects real-world model usage is a promising approach to bridging the gap between model development and deployment. Recent efforts, such as Copilot Arena (Chi et al., 2025) and WebDev Arena, have explored gamified arenas to gather data on human preferences, offering an alternative to purposefully curated datasets. However, such data collection methods may introduce noise, and arena-style approaches are not well-suited for long-horizon, interactive tasks. One potential approach is to leverage existing coding tools and environments, such as developing plugins for GitHub Copilot (Bajpai et al., 2024) or open-source IDEs, to capture real-world interactions. Unlike static datasets, human-centric data can also be collected encompassing diverse interaction modalities, such as users providing sketches to AI coding systems for web development (Li et al., 2024c). As AI coding systems continue to emerge and evolve, launching data initiatives focused on human-centric SWE data is also a crucial direction for advancing human-AI collaboration in software development.

### D.2. Training

#### D.2.1. ENVIRONMENT DESIGN FOR CODE RL

> Reinforcement Learning from Verifiable Rewards (RLVR) (Lambert et al., 2025) has emerged as a powerful paradigm in math and coding domains where model outputs can be evaluated against a ground truth outcome such as exact match and passing a set of unit-tests. Towards this direction, promising avenues include collecting executable codebase environments, sourcing task prompts/rewards from GitHub, and designing non-execution based rewards based on program syntax and semantics.
>
> *Challenges addressed: C.4, C.9*

**Collecting executable codebases:** In recent months, RLVR has seen success in solving algorithmic programming problems through DeepSeek-R1 (DeepSeek-AI et al., 2025) and OpenAI o1. Recently, on SWE-Bench, SWE-RL (Wei et al., 2025) use RL on a rule-based reward to improve performance on SWE-Bench. We find it promising to continue scaling the RL approach to problems collected from real-world software engineering repositories. Towards this, we believe that collecting execution-assisted gym-like reinforcement-learning environments will lead to further performance improvements. These environments can be used further to improve reasoning skills, environment-interaction capabilities and tool usage.

Several prior works (Jain et al., 2024c; Pan et al., 2024; Guo et al., 2025; Xie et al., 2025) curate executable environments for programming agents by supporting CI/heuristic-based repository installations. However, these works are at a relatively small scale and limited in scope, offering only a few thousand tasks from a maximum of a thousand repositories and more importantly, limited to the Python language. Scaling this up significantly requires solving several research and engineering problems. First, installing arbitrary repositories from Github, even using CI is challenging and we require smarter solutions potentially involving LLM-based installation agents. Next, setting up execution infrastructure would require storing installed repository images in something akin to `docker` for efficient storage and fast container startup times (Team et al., 2025). Notably, combined docker images can grow massively large and often grow at hundreds of gigabytes even at a modest scale of a few hundred repositories. They require engineering support for efficient storage and serving of such images.

**Sourcing task prompts and rewards:** Beyond environments, performing large-scale reinforcement learning would require collecting diverse challenging problems with an appropriate way to compute the rewards. These task prompts can be collected from Github (Pan et al., 2024) or generated synthetically from problems on Github. Moreover, assuming access to many executable repositories, we can source various end-to-end problems for tasks beyond bug-fixing such as optimization, fuzzing, etc. Access to pre-existing or generated test cases allows for measuring correctness and providing rewards.

However, we envision many practical challenges to remain. For example, longer-horizon tasks are usually more ambiguous and approaches may require multi-turn interactions beyond autonomous coding agents. This would pose a considerable

challenge during reinforcement learning where ambiguity resolution might need to be modeled in the reinforcement learning process itself. We elaborate on human collaboration further in Section D.2.3. Reward hacking (Skalse et al., 2022) poses another challenge as we build more real-world coding challenges. Test cases often suffer from coverage issues and can grade correct solutions as incorrect. For example, (Baker et al., 2025; Denison et al., 2024) identified that models attempt to bypass or cheat against the testing harness when optimized using reinforcement learning.

**Rewards without execution**: As setting up execution environments can lead to considerable overhead, another potential strategy is to use proxy metrics and trained language models to judge correctness. This was common in the pre-LLM era, researchers often used BLEU/CodeBLEU (Papineni et al., 2002; Ren et al., 2020) and BERTScore/CodeBERTScore (Zhang et al., 2019; Zhou et al., 2023) to assess correctness of text and code. In code, semantic and structural properties can be used to improve similarity metrics. Two examples of this are Dolos (Maertens et al., 2022), an AST-aware plagiarism detector, and `difflib.SequenceMatcher`, which can be used to compute the similarity between two patches (Wei et al., 2025; Ma et al., 2025b). Beyond rule-based rewards, LLMs-as-a-judge approaches can also be used as reward functions, possibly in conjunction with other execution-based or execution-free approaches.

### D.2.2. ADAPTING TO SPECIALIZED AND QUICKLY CHANGING CODEBASES

> Low-resource languages (Sec. C.7), custom APIs, library version updates (Sec. C.8), large codebases (Sec. C.5), and custom coding styles all surface the fact that code LMs struggle to adapt to unseen specialized contexts. Customization can be achieved through test-time training, keeping specialized information in an information bank. A cheaper and alternative approach to test-time training is to apply prompt and prefix tuning, where codebase-specific embeddings are learned and applied depending on the context.
>
> *Challenges addressed: C.7, C.8*

**Test-time training (TTT) to custom codebases**: TTT is the recent paradigm of adapting to a specific problem instance by training on a narrow set of in-distribution examples (Akyürek et al., 2024; Sun et al., 2020). This can be used when working in a low-resource context, for example training on a specific codebase, new domain, or unseen API. One challenge in this setting is customizing the model to the particular codebase while retaining general coding knowledge, potentially by using algorithms that can induce controllable forgetting (Wu et al., 2024). To get data in specialized contexts, we envision two mitigation strategies: generating synthetic data and collecting trajectories. In-distribution synthetic data can be generated in large quantities and then filtered and annotated with symbolic (e.g. compiler) information to gain a more global understanding of the current environment and setting. To gather agentic trajectories, we can keep track of previous model attempts and failures to learn from past successes and avoid making repeated mistakes. This will steer the model closer to the desired distribution–for example, to generate code in the specified version of libraries being used in the current context.

**Keeping an information bank of code information**: For library and versioning issues, retrieval (Sec. D.3.1) can be very effective for preventing hallucinations of wrong versions of libraries, which can inherently lead to better synthetic data and agentic trajectories. During the TTT process, we can also keep a large growing memory bank of code, documentation, synthetic code, and agentic trajectories in the specialized context. Retrieving from the memory bank would improve the success of generating code, which can then be augmented to the memory bank, and so on, continuously increasing the amount of data and knowledge available.

**Prompt and prefix tuning for specialized code contexts**: One issue that makes it difficult to continuously keep up with library updates is that doing full finetuning every time something changes is very expensive. Because only a small amount of knowledge needs to be learned compared to that of the pre-trained model, we believe less expensive approaches such as prompt tuning (Lester et al., 2021) and prefix-tuning (Li & Liang, 2021) could suffice. Both these methods append a set of learned task-specific vectors to the input sequence in order to model a specified context, though prompt tuning only modifies the input and prefix-tuning modifies the input at each layer. These methods have also been shown to have good OOD performance, and we believe they present a promising approach to dealing with multiple library versions. A separate prompt/prefix can be trained for each version and then applied according to the context. When an API has new updates, the prompt/prefix can then be cheaply re-tuned to reflect the new updates without undergoing full fine-tuning. This approach also applies to adhering to specific coding styles, where codebase-specific prompts/prefixes can also be learned.

**Learning on the fly**: When humans are faced with a task they have never seen before, they are often able to draw from past experiences and quickly adapt and generalize to the new domain. This is one of the big unsolved challenges of today's

LLMs: given an OOD coding task, how can models get up to speed and productively work on the task with few samples? On toy domains, an example of this is DreamCoder (Ellis et al., 2021), a system that learns to solve problems by writing programs and automatically discovering domain concepts and composing concepts together. Designing such approaches for more practical applications is an exciting research direction that will have drastic implications for coding and reasoning.

### D.2.3. TRAINING CODE LLMS TO COLLABORATE WITH HUMANS

> Training the next generation of code LLMs needs to account for human-AI collaboration, as these models will likely be deployed in ambiguous and interactive scenarios. We highlight two key directions for improving collaboration: First, learning to leverage specifications beyond natural language through formal methods and user-specified tests can mitigate vague specifications. Second, improving uncertainty quantification and proactive communication through post-training has the potential to prevent hallucination and misalignment.
>
> *Challenges addressed: C.3*

**Learning to Leverage Specifications Beyond Natural Language**: As discussed in Section C.3, while natural language prompts offer intuitive and flexible ways to express requirements, they often suffer from ambiguity and incompleteness. One direction to address this limitation is to train models to leverage enhanced specifications with more precise and verifiable representations, such as formal specifications and test-based specifications.

*Formal specifications*: To mitigate underspecification issues, one solution is to develop systems that can translate user intent into formal specifications (Szegedy, 2020; Endres et al., 2024). While current autoformalization approaches face challenges in accurately capturing user intent (see example below), we envision next-generation systems that will iteratively refine formal specifications through interactive verification with human feedback. These systems would present intermediate formalizations in accessible notation, enabling non-expert users to verify correctness before code generation.

> *Example: Incomplete specification in Verus*: Here, we show a failure mode of LLMs when writing specifications and proofs in Verus. The LLM is asked to write the `ensures` postcondition clause for a a ring buffer enqueue function[a]. Here, the postcondition is incomplete: it does not check, for example, that the original elements were maintained in the ring buffer.
>
> ```
> fn enqueue(&mut self, val: T) -> (ret: bool)
>
>     ensures
>         ret == !old(self).is_full(),
>         self.inv(),
>         if ret { self.view() === old(self).view().push(val) } else { self.view() ===
>     ↪ old(self).view() }
>
> {
>     if self.is_full() {
>         false
>     } else {
>         self.ring.set(self.tail, val);
>         self.tail = (self.tail + 1) % self.ring.len();
>         true
>     }
> }
> ```
>
> ―――――――――
> [a]Full example here

*Tests as specifications*: Another approach to specify software behavior is through tests. These range from input-output examples and assertions to property-based tests. However, in practice, hand-crafted test suites are often incomplete, failing to capture the full intended behavior, particularly edge cases. This can lead to misalignment, where AI-generated code passes tests but does not genuinely meet functional requirements, potentially misleading users. Moving forward, a direction is training models to generate high-quality test cases based on the user's initial query, ensuring more comprehensive specification coverage.

> *Example*: For instance, in a release of AI CUDA Engineer by Sakana AI, an AI-generated CUDA kernel for lower triangular matrix multiplication—purportedly achieving significant speedups—was later found to exploit out-of-bounds memory access to bypass correctness checks[a]. Advancing research on frameworks that facilitate test generation and automated adversarial testing represents an important direction.
>
> ---
> [a]The full LLM-generated kernel code can be found in Listing 3, pg. 46-47 of (Lange et al., 2025)

**Learning to Quantify Uncertainty and Communicate Proactively**: As AI coding systems are increasingly deployed to complex software engineering tasks, they encounter more ambiguous and uncertain scenarios compared to traditional benchmarks for coding models. Ideally, in such situations, these systems should proactively communicate with users to clarify tasks and acknowledge its own limitations rather than becoming stuck in endless failure loops or generating buggy code. A key challenge is enabling models to distinguish between well-specified and ambiguous instructions while quantifying uncertainty in a robust manner. While early studies, such as Vijayvargiya et al. (2025) and the example below, demonstrate that interactive LLMs can improve performance through clarification-seeking behavior, current models still struggle with uncertainty estimation. Equipping models with the ability to quantify uncertainty will likely require incorporating corresponding reasoning data into the post-training stage.

Besides uncertainty quantification, Shao et al. (2024b) identify communication as a primary challenge in human-agent collaboration, highlighting the need for improving models' proactive communication capability. Current models often fail to ask meaningful questions when user input is ambiguous or insufficient, and they struggle to provide progress updates or verify plans in interactive settings. Enhancing models' proactive communication abilities requires innovative approaches to reward behaviors that yield benefits over multiple steps. Since communication with users does not immediately resolve the task at hand but may improve long-term outcomes, effective strategies must account for delayed rewards in training.

> *Example: Discussion Helps Coding Agents Resolve Github Issues:* In SWE-bench (Jimenez et al., 2024) `pydata_xarray-4750`, the original issue description requests limiting the number of data rows displayed in `repr`. While it suggests a maximum of 25 rows, it does not specify whether this number should be configurable—a key requirement that emerged during the issue discussion. When SWE-Agent (Yang et al., 2024b), powered by GPT-4o, uses only the issue description as the problem statement, it generates a function that hardcodes the maximum at 25, causing the solution to fail the test. However, incorporating the issue discussion allows the agent to produce a correct, test-passing implementation (see Listing 2). This suggests that enabling coding agents to engage in discussions with users could potentially improve the issue solving rate.

### D.3. Inference Time Approaches

#### D.3.1. SEMANTIC-AWARE EMBEDDINGS AND RETRIEVAL

> In contrast to text, embeddings for code should incorporate execution and semantic information, improving retrieval. RAG benefits from both context-aware retrievals and explicit training on how to use them, enhancing code reuse across languages and APIs. Beyond static retrieval, AI agents could also dynamically navigate codebases using command-line tools and IDE functions.
>
> *Challenges addressed: C.5*

**Semantic and execution aware code embeddings**: When training LLMs, code is often treated as pure tokens (just like text) rather than explicitly incorporating code-specific information such as program execution and semantics. As a result, code that is close in embedding space is more often syntactically similar than semantically similar (Utpala et al., 2023; Zhao et al., 2023), and there are few reliable methods today to retrieve semantically similar code. However, before the LLM era, there were a variety of efforts to incorporate code properties when training embeddings. For example, Nye et al. (2020) train neural modules to represent program operations, leading to compositional program representations that encode the semantics of the underlying programming language. Many other works (Zohar & Wolf, 2018; Ellis et al., 2019; Chen et al., 2021b) attempt to learn execution-aware latent representations for partial and full programs, taking semantics into account.

We speculate that incorporating these techniques to train models to have better and more semantically aware representations

*Listing 2.* SWE-Agent improves when incorporating issue discussions

```
# Use only the issue description
def _mapping_repr(mapping, title, summarizer, col_width=None):
    if col_width is None:
        col_width = _calculate_col_width(mapping)
        summary = [f"{title}:"]
    if not mapping:
        summary += [EMPTY_REPR]
    else:
        items = list(mapping.items())
        if len(items) > 25:  # Show first 12 and last 12 items
            first = items[:12]
            last = items[-12:]
            summary += [summarizer(k, v, col_width) for k, v in first]
            summary += ["    ..."]
            summary += [summarizer(k, v, col_width) for k, v in last]
        else:
            summary += [summarizer(k, v, col_width) for k, v in items]
    return "\n".join(summary)

# Use the issue description and discussion
def _mapping_repr(mapping, title, summarizer, col_width=None):
    col_width = _calculate_col_width(mapping)
    summary = [f"{title}:"]
    if mapping:
        items = list(mapping.items())
        if title == "Data variables" and len(items) > OPTIONS["display_max_rows"]:
            # Show first and last variables if there are too many
            first_n = OPTIONS["display_max_rows"] // 2
            last_n = OPTIONS["display_max_rows"] - first_n
            selected_items = items[:first_n] + [("...", "...")] + items[-last_n:]
        else:
            selected_items = items
        summary += [summarizer(k, v, col_width) if k != "..." else "    ..."
            for k, v in selected_items]
    else:
        summary += [EMPTY_REPR]
    return "\n".join(summary)
```

may lead to models with a more general understanding of code (Sec. C.6). For example, if correct and buggy programs could hypothetically be separated in embedding space, then models could be steered away from the incorrect program space. While such a clean separation might not be possible, we believe that training embeddings to have interesting semantic properties is worth exploring.

**Better retrieval-augmented code generation**: When retrieval-augmented language models were first introduced, they often relied on training the retriever and language model jointly, as in FiD (Izacard & Grave, 2020), RETRO (Borgeaud et al., 2022), and Atlas (Izacard et al., 2023). As language models increased in size, the field shifted to a black-box setting (Shi et al., 2023), where the retrieval module is tuned independently to adapt to the pretrained black-box LLM. This setting is much more cost-effective, but the language model is not explicitly trained on how to use its retrievals.

The black-box setting is ideal for challenges such as low-resource languages or specialized contexts. In these situations, the model has not seen enough training data to fully grasp the context, and the challenge is often syntactic rather than algorithmic. For example, when adapting to a domain or a codebase where the relevant API functionality or code style, retrievals can be very instructive. When using APIs with multiple versions, providing retrievals in the correct version can inform the model of how to use the API. When writing code in a completely new language, showing examples of `for` loops and `while` statements will teach the model the syntax of these constructs. Retrievals should be diverse and given in multiple forms, including documentation, function definitions of APIs that are used, and example use cases of target functions.

In many other cases, however, we believe that a black-box setting is insufficient. As described in Sec. C.5, there are two challenges: 1) knowing what to retrieve and 2) using the retrieval. The first challenge relies on retrieving relevant examples, both syntactically and semantically. We believe that having more semantically aware embeddings, as mentioned above, will drastically improve this. For example, embeddings can be trained contrastively to minimize the distance between semantically similar programs. Another potential direction is to consider a diverse set of potential retrievals and then train the retriever to prefer samples that help during generation, as in Atlas (Izacard et al., 2023).

The second challenge, using the retrieval, is a code reuse task, which requires complex reasoning and code understanding. Algorithms provided in retrievals may often need to be modified and adapted significantly to adapt to the current setting. An example of this might be writing a C++ version of a shortest path algorithm when the retrieval is a Java version, a translation task that models may not have been trained for explicitly. Long chunks of retrieved documentation may need to be understood precisely so that correct hyperparameters and flags can be used. Yet, in a black-box setting, models have not been explicitly trained to leverage this information. Therefore, just as training on incorrect-correct code pairs can improve program repair, we believe that direct training can be very beneficial for code reuse and retrieval-augmented generation. Execution information could also be useful, as code reuse often requires understanding the situation well enough to identify subtle differences between the context of the retrieved code and the current context.

**Retrieving via code navigation on the fly**: Standard retrieval-augmented methods keep a large retrieval index containing millions of embeddings, which can require a high one-time cost to create. As the codebase evolves, these embeddings may also need to be continuously updated. Instead of keeping track of embeddings, another approach is to find retrievals on the fly by navigating the codebase. We can imagine an agent that learns to use command line functions such as `cd`, `ls`, and `grep`, as well as IDE functions such as jumping to function definitions or finding all references of a function. Static analysis tools can also be paired with the agent to improve code navigation, such as providing the abstract syntax tree (AST) or file structure of a codebase.

### D.3.2. INTEGRATION WITH SWE DEVELOPMENT FRAMEWORKS

> Integrating AI with SWE development frameworks is critical for practical applications and impact on developer workflows. While software development is inherently integrated with tools, workflows, scaffolding, and meta-code, these are often absent from source code and scarce in AI training data. Ensuring that AI deeply understands software deployment beyond code editing is crucial, as writing code is only a small part of the development cycle. These can include automated reviews, deployment risk assessments, and documentation generation. We can also fine-tune LLMs to recognize and avoid known software anti-patterns such as CWEs.
>
> *Challenges addressed: C.4, C.5*

**Incorporating AI into the CI/CD process**: In continuous integration and continuous deployment (CI/CD), automated pipelines are the backbone for building, testing, and deploying code changes. CI/CD accelerates feedback cycles and minimizes integration issues. AI offers several integration points within CI/CD. AI-powered code review tools can be incorporated into CI pipelines to automatically identify and flag style violations, potential security vulnerabilities, and code smells before human reviewers are involved. Furthermore, AI can provide intelligent deployment risk assessments. By analyzing code changes, test outcomes, and historical deployment data, AI can predict the likelihood of deployment issues, informing decisions about whether to proceed with automated deployment or mandate manual verification steps. Finally, AI can automate the generation of release notes by summarizing commit messages, issue tracker data, and relevant code modifications within the CI/CD process.

**Steering away from software anti-patterns**: In software engineering, certain anti-patterns frequently lead to bugs. For example, common weakness enumeration (CWE) is a categorization of software and hardware weaknesses often leading to vulnerabilities. Because publicly available GitHub code often contains code with anti-patterns, bugs, and CWE vulnerabilities, LLMs often write code susceptible to these issues (Asare et al., 2023; Fu et al., 2023). We hypothesize that explicitly steering models against these vulnerabilities will lead to more secure and correct code. One way to do this is to collect a large number of program samples violating each CWE (either synthetically or on GitHub) and then use these samples as negative signal during further supervised fine-tuning or RL stages.

D.3.3. INCORPORATING SWE TOOLS

> Software engineers integrate a variety of domain-specific tools when writing code. By repeatedly interacting with tools in an RL-style manner, AI can develop the ability to do the same. Beyond tool use, using neurosymbolic approaches such as incorporating program analysis and type-checking can also help enhance LLM capabilities.
>
> *Challenges addressed: C.2*

**Learning to use SWE Tools**: As mentioned in Sec. C.2, we believe SWE agents should understand the intricacies of programming tools and be able to autonomously invoke them as needed. There are three skills to learn: which tool to use, how to use the tool, and how to incorporate the results of the tool. Similar to how models learn to play complicated games, we believe that intelligent tool integration can be learned through repeated interactions with the tool in a RL-style manner. One way we envision this is as follows: first, the interface of the tool must be precisely specified. Next, data containing repeated interactions from the tool (with varying degrees of success) should be collected. Finally, multiple rounds of RL and expert iteration can be done to improve understanding of the tool and learn from misuses.

Evidence that learning higher-level strategies might be possible is that through test-time techniques, OpenAI's o3 model learned to write brute-force solutions to verify the correctness of more complicated solutions (El-Kishky et al., 2025). We envision that after learning to use tools, AI coding agents can autonomously invoke tools as needed to improve its overall world model of the code and hence its software engineering capabilities.

**Neurosymbolic Approaches**: Code is a unique domain because there is a vast body of techniques from programming languages (PL) research to build off of, but the majority of AI for code research today does not leverage the symbolic properties of code. Some of these PL techniques are as follows: abstract interpretation (Cousot & Cousot, 1977) is a technique to compute over-approximations of program state in order to prove the soundness of program properties at points in the code. Concolic testing (Godefroid et al., 2005; Sen et al., 2005) finds bugs in software by combining concrete and symbolic execution. Model checking (Clarke, 1997) is a way to prove properties of execution traces via temporal logic. Linting and type-checking (Cardelli, 1996) provide a static check to ensure that variables, expressions, and functions adhere to a programming language's rules. Finally, many other program analysis algorithms leveraging these tools have been designed to prevent bugs and ensure code correctness properties.

Traditional PL approaches have a few common shortcomings, which overlap with some of the issues mentioned in Sec. B.6. First, they often require very complete and precise specifications. Many tools need to have specifications for all library functions, need to specialize to a precise version of the language, and need to specialize to the build system. Second, there is often a high computational cost due to the large search space. Third, there can be many false positives due to the limitations of the tool. We believe that deeply integrating these symbolic tools with LLMs can partially mitigate these challenges.

We provide a few examples of this potential integration. When generating code, program analysis techniques could be applied on shorter snippets of AI-generated code to surface potential bugs or prove properties of the generated code. To improve general code understanding, LLMs can be trained with information about program structure such as abstract syntax trees (Gong et al., 2024). When debugging a large codebase, when the scale is too large to directly apply PL techniques, AI could be first used to narrow down potentially problematic sections of the code which are then handed off to PL tools for debugging. During code generation in DSLs, LLMs can leverage the grammar of the programming language to do constrained decoding (Poesia et al., 2022; Geng et al., 2023; Wei et al., 2023b) to mitigate syntactic errors. During code refactoring, abstract interpretation and static analysis can be used to identify whether new errors have been introduced and preemptively cut off unpromising search paths.

**Deductive Synthesis and Intermediate Languages**: Early program synthesis relied on *deductive synthesis* approaches (Burstall & Darlington, 1977), where programmers would write a clean simple implementation and then apply transformation rules to convert it into a more efficient one. The appeal of deductive approaches is that because these rewrite rules are semantics preserving, there is a correct-by-construction guarantee. One success story of deductive synthesis is Spiral (Puschel et al., 2005), a DSL for signal processing kernels that takes advantage of domain-specific transformation rules to produce implementations beating expert hand-crafted code. Another example is Halide (Ragan-Kelley et al., 2013), a DSL for high-performance image and array processing code. Due to the difficulty of writing optimized code, humans generally opt for writing code in these intermediate DSLs, and we find it promising for LLMs to do the same.

> *Example. LLM-aided Compilation for Tensor Accelerators*: As an example, Hong et al. (2024) consider the task of generating highly optimized, hardware-efficient code for a tensor accelerator from a high-level specification of a computation (e.g. C++ code). Their pipeline works in two steps: first, the high-level specification is translated to a DSL. Then, the DSL code is symbolically compiled to hardware-specific instructions. The LLM is also used to optimize the DSL code via a cost model driven search, where it suggests rewrites and scheduling operations (e.g. loop reordering) that guarantee semantic equivalence.

### D.3.4. SCAFFOLDING HUMAN SUPERVISION

> At inference time, most machine-generated code will be presented to humans in a format shaped by the human-AI interface design. Since AI may be responsible for generating the majority of the code within a human-AI team, it is important to ensure human control and oversight. By scaffolding human supervision with techniques like summarization and interactive verification, we could potentially improve trust in AI-generated code.
>
> *Challenges addressed: C.3*

Once code LLMs are deployed for inference, it is crucial to scaffold human supervision of AI-generated code. This goes beyond merely enhancing the accuracy of AI-generated code, as humans often still need to make the final decision on whether to accept the code or understand it for future integration and maintenance. A study on Github Copilot usage (Al Madi, 2023) revealed that programmers tend to allocate less visual attention to AI-generated code. While one solution is to train humans to better identify issues in AI-generated code (Singhal & Kumar, 2023), a more desirable approach is to design AI systems that scaffold human supervision, reducing their cognitive load when reviewing generated code.

One way to achieve this is by enriching AI-generated content with additional contextual information. Modern LLM chatbots now routinely generate text with citations for knowledge-intensive queries. In Collaborative STORM (Jiang et al., 2024), researchers demonstrated that dynamically presenting hierarchical "mind maps" alongside the actual collected information significantly enhanced human-AI collaboration, particularly in long sessions. In software engineering specifically, Sun et al. (2024b) highlighted the benefits of high-quality source code summarization in aiding software developers in understanding and maintaining machine-generated code. Second, interactive approaches can also enhance supervision. One example is *Live Programming* (Ferdowsi et al., 2024), a continuous display of the runtime values of a program, as a means of lowering the cost of validating AI-generated code. However, these existing studies are largely limited to specific programming languages and small codebases. Finally, improving the readability and interpretability of AI-generated code itself presents a promising direction. For example, Pu et al. (2020) showed that modeling program synthesis as rational communication improved end-user interpretation and subsequent communication of code. Expanding on these ideas, future research should prioritize human interpretability in the design and optimization of AI coding systems, fostering greater trust and control in AI-assisted software development.

## E. Limitations

We identify a few limitations below:

**Speculative nature of future work**: The ideas we list in the future work section are opinionated directions we believe have a high chance of success. Many draw upon insights from related work in the literature, but many lack strong and concrete evidence. We encourage further research validating or disproving the effectiveness of these ideas.

**Limited scope of future work**: We also do not include any novel moonshot ideas, and many of the directions we propose have their roots in existing code LLM literature. Our future work section is also relatively general and applies holistically to AI for code. However, the field has many tasks and challenges that can benefit from using domain-specific knowledge and insights, and we do not touch on these. Finally, this paper is written by people primarily in the academic community, who may not know the details of cutting-edge methods employed in frontier industry labs. We cater this paper towards areas we have more expertise in, and thus leave out many promising directions such as novel architectures.

**Focus towards code-specific challenges**: In this paper, we mostly focus on code-specific challenges and techniques. However, there are many techniques that apply to general LLM reasoning and development that could be directly applied to code. We believe many of these methods can be used in synergy with code-specific techniques.

**Quickly changing nature of the field**: The field of LLM for software engineering is progressing very rapidly, with new innovations released weekly. It is possible that a reader reading this paper a few months down the line will find that several of the mentioned challenges will have been partially or entirely resolved.

## F. Conclusion

In this position paper, we have identified key tasks at the heart of AI for software engineering as well as a set of three measures to classify different realizations of these tasks. We have also highlighted critical cross-cutting challenges that permeate throughout many tasks. Finally, to drive progress in AI for code, we've pinpointed a set of exciting and promising research directions for alleviating these challenges and advancing AI towards being a more capable software engineer. We hope this work provides valuable insights about the current landscape of AI for software engineering and encourages future research in these directions. By building on these insights, we are optimistic that the community can work toward developing AI-driven solutions that better support software engineers in real-world settings.

## G. Acknowledgements

We thank Alex Polozov, Baptiste Roziere, Daya Guo, Jenny Liang, Jiawei Liu, Justin Chiu, Kexun Zhang, Leonardo Hernandez Cano, Li Zhong, Michael Wang, Silas Alberti, Theo Olausson, Valerie Chen, Xingyao Wang, Yangruibo Ding, Yuxiang Wei, Zhiruo Wang, and several anonymous workshop reviewers for providing valuable feedback regarding various stages of the draft.

We also thank the following people for bringing up illustrative examples mentioned in this paper: Silas Alberti (debugging cloud applications), Chuyue Sun (incomplete specification in Verus), MIT's 6.172 Course (performance instrumentation), Theo Olausson (costly disasters), Songlin Yang (syntax error in Triton).

A. Gu is supported by the National Science Foundation (NSF) Graduate Research Fellowship under Grant No. 2141064. N. Jain is supported by NSF grants CCF:1900968, CCF:1908870, and by SKY Lab industrial sponsors and affiliates. A. Solar-Lezama is supported by the National Science Foundation (NSF) and Intel Corporation through NSF Grant CCF:2217064. D. Yang is supported by the ONR YIP Award N000142412532.

## H. Survey of Related Work: Tasks in AI Software Engineering

In this section, we briefly survey some of the relevant works for each of the tasks we mention in Sec. B. These works are by no means complete, and we encourage the reader to check out the survey works mentioned in the introduction and in this section for further references.

### H.1. Code Generation

**Code Completion**: Completion typically happens in conjunction with live programming or within an IDE, helping developers write code faster by suggesting relevant continuations. Traditional code completion systems rely heavily on syntactic and type-aware models (e.g., AST-based models), but recent advances leverage LLMs trained on code corpora to offer semantically rich and context-aware suggestions, naturally following the next-token prediction task in language modeling (Radford et al., 2019). Tools like GitHub Copilot and Codex exemplify this trend (Chen et al., 2021a), and are followed by commercial tools such as Cursor[5] and Tabnine[6]. Recent advances in context-aware (Agrawal et al., 2023), grammar-aligned (Park et al., 2024), and constraint-based decoding (Sun et al., 2023) have improved the quality of local completions, particularly for shorter code snippets. For longer code snippets, the typical task formulation is method implementation synthesis given a function signature. This setup is commonly evaluated using benchmarks such as MBPP (Austin et al., 2021) and HumanEval (Chen et al., 2021a).

**Natural Language to Code Generation**: Translating natural language into code has long been a central challenge in AI for programming. Early attempts at code generation involved semantic parsing (Zettlemoyer & Collins, 2012; Wong & Mooney, 2006), where natural language is translated into logical forms or domain-specific languages. A prominent example is SQL query synthesis from natural language questions, as seen in systems like Seq2SQL (Zhong et al., 2017) and

---

[5]https://www.cursor.so
[6]https://www.tabnine.com

Spider (Yu et al., 2019), where the target language is constrained, small, and domain-specific. Recent work demonstrates that large language models (LLMs) can generalize to general-purpose programming languages, enabling the generation of larger and more complex code snippets (OpenAI, 2023b). When applied to code completion, users often begin with natural language instructions in the form of comments, which LLMs use as context for code synthesis. Beyond function-level code generation (Austin et al., 2021; Chen et al., 2021a), recent work has extended to class-level generation (Du et al., 2023), which targets classes in object-oriented programming, and even project-level code generation (Cao et al., 2024; Wang et al., 2024f), which involves generating or completing entire multi-file codebases.

**Multimodal Code Generation**: While text can describe most cases of code generation, certain instructions are better defined visually. For example, in graphics applications, visual context such as a trajectory or a 3D model is essential to synthesize the correct code. Demonstrations of GPT-4's multi-modal capabilities have shown that models can generate functional webpage code directly from paper sketches, translating visual layouts into HTML and CSS (OpenAI, 2023a). LogoMotion (Liu et al., 2025b) explores visually grounded code synthesis for animations and motion graphics in JavaScript. The system leverages vision-language models (VLMs) to incorporate both visual inputs and user instructions, enabling code generation that aligns with spatial and temporal visual cues. Other works, such as SynthesizeCAD (Nandi et al., 2020) and SGP-Bench (Qiu et al., 2024), explore how LLMs can interface with visual and 3D modalities by generating code in languages like SVG and CAD.

**Code Generation in Low-Resource Languages**: As discussed in Sec. C.7, one major challenge is writing code in low-adoption general purposed language and domain specific languages (DSLs). Benchmarks for this include MultiPL-E (Cassano et al., 2023), McEval (Chai et al., 2024), and VerilogEval (Liu et al., 2023b). A popular method to improve performance is to train on manually curated and processing data in low-resource languages such as Coq (Florath, 2024) and Verilog (Pei et al., 2024). Another line of work aims to achieve transfer between different low-resource languages (Paul et al., 2024; Cassano et al., 2024; Orlanski et al., 2023). Finally, since the lack of data is a large bottleneck, another popular direction is using relevant retrievals such as useful functions and library documentation (Yang et al., 2023b; Zhou et al., 2022; Yang et al., 2023b). For a recent survey of code generation for low-resource languages and DSLs, see (Joel et al., 2024).

**Security Concerns Surrounding Code Generation**: Despite the growing power of LLMs for code generation, their outputs often remain insecure, incorrect, or misaligned with user intent. For instance, BaxBench (Vero et al., 2025) evaluates LLMs on generating secure and correct back-ends, revealing that while the average functional correctness is already modest ($\sim 60\%$), the rate of secure outputs is even lower ($< 35\%$). To better understand and quantify these limitations, several benchmarks and evaluation suites have been proposed. SecurityEval (Siddiq & Santos, 2022), SafeCoder (He et al., 2024), CodeLMSec (Hajipour et al., 2023), CWEval (Peng et al., 2025), and CyberSecEval (Bhatt et al., 2023; Wan et al., 2024a) each provide distinct lenses on evaluating vulnerabilities, unsafe API usage, or compliance with common weakness enumerations (CWEs). In response, several approaches introduce human-in-the-loop guardrails, where developers can interactively guide, inspect, or constrain the generation process. Dynex (Ma et al., 2025a), for instance, supports dynamic, step-wise code synthesis with user feedback, enabling real-time correction and iterative refinement before errors can accumulate.

**Human Interaction in Code Generation**: Modern code LLMs typically support interactive code generation through conversational interfaces. Champa et al. (2024) conducted a quantitative analysis of developer-ChatGPT interactions using the DevGPT dataset (Xiao et al., 2024), examining how the quality of the initial prompt influences conversation length. Code LLMs can be further optimized for various interactive scenarios, including debugging environments (Surameery & Shakor, 2023), educational settings (Kazemitabaar et al., 2023a;b; Prather et al., 2023; Sheese et al., 2024), and use by non-professional programmers (Yan et al., 2024). Beyond human-driven interactions in chat-based setups, more advanced code generation systems such as coding agents can proactively ask clarifying questions (Vijayvargiya et al., 2025) or generate test cases for users to validate (Lahiri et al., 2022; Fakhoury et al., 2024) before generating the actual code, helping to resolve ambiguities.

## H.2. Code Transformation

**Code Refactoring**: Code refactoring aims to simplify and remove repetitions in complex repositories without altering high-level program intent. While there have been traditional methods (Pailoor et al., 2024) that refactor data structures, Aider AI introduces a refactoring benchmark[7] evaluating LLM's ability to output long chunks of code that simplify complex

---

[7] https://github.com/Aider-AI/refactor-benchmark

programs without changing its behavior. More recently, RefactorBench (Gautam et al., 2024) introduced a more complex benchmark with natural language refactor requests, as well as an LLM agent that can perform refactoring.

**Code Migration**: Compared to code refactoring, code migration typically refers to mid-scale modifications that affect a program's interface, dependencies, or underlying architecture. Common examples include switching the back-end database from MySQL to PostgreSQL, migrating a machine learning model from TensorFlow to PyTorch, or upgrading the Java version from legacy Java 8 to a more modern Java 17. While recent work has introduced benchmark designed to evaluate library migrations (Islam et al., 2023), works at Google (Nikolov et al., 2025) and Amazon (Omidvar Tehrani & Anubhai, 2024) have explored LLM-driven solutions for simple but vast migrations. Google's system identifies locations for changes, generates edits with LLMs fine-tuned on internal code, and automatically validates changes through compilation and test execution.

**Code Translation (Transpilation)**: Moving beyond code migration, transpilation involves large-scale transformation of a program's underlying programming language. Transpilation serves not only to modernize outdated codebases but also to eliminate classes of safety issues inherent to older languages. A particularly active area of research involves transpiling C-based systems to Rust, a systems-level language that offers strong memory and concurrency safety guarantees. This direction has garnered attention, including from the U.S. Department of Defense[8], which maintains critical infrastructure built on aging C code. An end-to-end LLM-based approach, such as Flourine (Eniser et al., 2024), has been proposed for real-world code translation, but it has achieved only limited success due to frequent compilation errors. Recent efforts like Syzygy (Shetty et al., 2024), C2SaferRust (Nitin et al., 2025), and AlphaTrans (Ibrahimzada et al., 2024) have shown the potential for hybrid approaches combining LLMs with traditional program analysis techniques. However, some significant challenges remain, as identified by Li et al. (2025b), including ensuring correctness in large codebases while maintaining desirable attributes such as speed, reduced vulnerabilities, and idiomaticity. Specifically, We anticipate that the techniques discussed in Section H.3 may help address these remaining challenges.

**Code Optimization:** Certain refactoring or transpilation tasks are specifically aimed at optimizing code performance. Prior work has explored the use of LLMs for optimizing standalone programs, such as PIE (Shypula et al., 2023), which targets C++ functions, and AlphaDev (Mankowitz et al., 2023a), which discovers more efficient sorting algorithms at the assembly level. These tasks are particularly challenging due to the vast search space of possible code transformations. More recently, KernelBench (Ouyang et al., 2025) introduced a benchmark focused on optimizing machine learning models written in high-level PyTorch code into low-level, high-performance CUDA GPU kernels. For a broader overview of language models applied to code optimization, see the survey by Gong et al. (2025).

### H.3. Software Testing and Program Analysis

**Short-horizon Testing**: For short-horizon testing such as unit tests (Lemieux et al., 2023) and property-based tests (Vikram et al., 2023), LLMs are employed to automatically generate targeted test cases (Li & Yuan, 2024; Mündler et al., 2025), and even hill-climb on code coverage to improve test effectiveness (Ryan et al., 2024). At the granularity of individual functions, LLM-generated tests have also been employed to support downstream tasks such as filtering implementations based on behavioral correctness (Chen et al., 2022; Zhang et al., 2023b), as well as assisting in program debugging by surfacing inputs that expose incorrect behavior (Chen et al., 2025).

**Long-horizon Testing**: Long-horizon testing involves evaluating system behavior across extended executions, complex interactions, or multiple components, potentially embedded within a CI/CD (Continuous Integration or Delivery) pipeline. Fuzzing (Miller et al., 1990) is a long-horizon testing approach that continuously generates novel random input. Recent works such as Fuzz4All (Xia et al., 2024b), KernelGPT (Yang et al., 2023a), and OSS-Fuzz (Liu et al., 2023a; Chang et al., 2024) have shown that LLMs can significantly improve effectiveness through better input generation and exploration strategies. Specificatlly, OSS-Fuzz-Gen (Liu et al., 2024b) employs diverse LLMs for fuzzing harness generation, helping to find novel and complex crashing interactions.

**Static Analysis for Vulnerability Detection**: Vulnerability Detection refers to the task of identifying weaknesses or flaws in software code that could be exploited to compromise the system's security, stability, or correctness. A wide range of prior work leverages machine learning models such as Graph Neural Networks (GNNs) and Recurrent Neural Networks (RNNs) to detect software vulnerabilities (Zhou et al., 2019; Chakraborty et al., 2020; Dinella et al., 2020; Hin et al., 2022; Li et al., 2021). While some recent methods pre-train or fine-tune LLMs on code-specific datasets (Fu & Tantithamthavorn,

---

[8]https://www.darpa.mil/news/2024/memory-safety-vulnerabilities

2022; Steenhoek et al., 2023; Cheng et al., 2022) to improve vulnerability classification, several studies have highlighted the limitations of LLMs in real-world software (Steenhoek et al., 2024; Ding et al., 2024a; Khare et al., 2023). To combat such limitations, works like Li et al. (2024a), IRIS (Li et al., 2024e), LLMDFA (Wang et al., 2024b), and InferROI (Wang et al., 2023) explored augmenting static analysis tools (e.g., CodeQL) with LLMs for taint and resource leak analyses. More recently, BigSleep (2024) demonstrated the potential of using LLMs at a much bigger scale by finding a real SQLite vulnerability through exploratory variant analysis.

**Specialized Program Analysis**: Beyond long-running analysis to identify vulnerabilities, several traditional program analyses have struggled to scale in practice despite their theoretical promise. For instance, inferring program invariants (properties deemed to always be true at a program point) has been challenging with traditional symbolic methods such as Daikon (Ernst et al., 2007; Padon et al., 2016) while being valuable for exposing bugs (Hangal & Lam, 2002) and aiding software evolution (Ernst et al., 1999). Similarly, type inference for dynamically typed languages suffers from coverage limitations of rule-based approaches and requires specialized tools like ShapeIt (Zheng & Sen, 2024) for domain-specific challenges such as inferring symbolic tensor shapes.

**Specification Inference**: Specification inference is the task of automatically recovering formal description of a program's expected behavior, including pre-conditions, post-conditions, or invariants. The availability of specification is at the core of establishing *trust* (Roychoudhury et al., 2025b), and existing works (Dinella et al., 2024b; Ruan et al., 2024) have shown that LLMs can help the inference of such specifications. For instance, Dinella et al. (2024a) presents a program structure aware technique for synthesizing pre-conditions for arbitrary code snippets, and have established a dataset of 18K LLM generated pre-conditions on real Java projects.

**Invariant Inference**: As a subtask of specification inference, invariant inference aims at inferring loop, function, or class invariants, which are greatly helpful in automatic program verification. There have been several LLM-based approaches for invariant identification. They enhance traditional approaches through structured representations (Si et al., 2018), LLM-based prompting (Kamath et al., 2023; Pei et al., 2023) and re-ranking (Chakraborty et al., 2023), and reinforcement learning (Yu et al., 2023). Similarly, works have used sequence-to-sequence models (Wei et al., 2023a), few-shot LLM approaches like TypeGen (Peng et al., 2023), and generate-then-rank methods like TIGER (Wang et al., 2024a) for type inference. Consequently, we observe new benchmarks emerging in the space such as LIG-MM (Liu et al., 2024a) for loop-invariant detection.

**Binary Analysis**: While the aforementioned tasks primarily focus on human-readable programming languages, many can also be extended to operate on compiled machine code, or binaries. One prominent example is binary type inference, which aims to recover high-level type information from low-level binary code. It has seen significant improvements with deep learning models and LLMs (Pei et al., 2021; Zhu et al., 2024). These advancements, alongside other LLM-based analyses, have enhanced the capabilities of decompilers, enabling them to synthesize human-readable code from binaries (Liu et al., 2025a). Beyond decompilation, LLMs have also been applied to detect security vulnerabilities in binaries (Liu et al., 2023c) and to generate semantic summaries that capture the high-level intent of binary code (Jin et al., 2023).

### H.4. Software Maintenance

**Code Navigation**: Code navigation refers to the task of locating a specific position within a code repository based on either a natural language description (Liu et al., 2024f) or a programmatic specification (Avgustinov et al., 2016). Common use cases include identifying where a particular functionality is implemented, tracing the origin of user input that leads to a vulnerability, or locating relevant files when starting work on a new feature. This capability underpins many downstream tasks such as software testing, vulnerability detection, program repair, and code question answering. Code navigation or code search modules are integral components of modern code agents (Yang et al., 2024b; Bouzenia et al., 2024; Xia et al., 2024a), often implemented using find commands, embedding-based similarity search, or query-based tools like CodeQL and Semgrep.

**Code Documentation and Summarization**: Several works have used LLMs for code summarization invoking techniques like prompting (Sun et al., 2024b; Su & McMillan, 2024; Haldar & Hockenmaier, 2024; Ahmed et al., 2024b). RepoAgent (Luo et al., 2024) is a framework that analyzes global contextual relationships in source code to generate fine-grained documentation. Shi et al. (2024) show that LMs are capable of generating good natural language outlines – text descriptions alongside code to partition it into semantically coherent sections. One challenge is that the evaluation of this task is very tricky: the academic community currently lacks datasets and benchmarks that contain good documentation and the automatic evaluation metrics do not align well with human metrics (Diggs et al., 2024).

**Pull Request (PR) Review**: In industry, autonomous software agents such as OpenHands (Wang et al., 2024g) and Devin have been able to automatically review and even fix PRs. At ByteDance, BitsAI-CR (Sun et al., 2025) is a code review system that identifies issues based on a manually crafted taxonomy of review rules. In the academic community, there have been several works studying the ability of AI systems to automatically review PRs (Tufano et al., 2021; 2022; Li et al., 2022b; 2024b). Recently, AutoCodeRover (Zhang et al., 2024b) combines LLMs with code search to automatically fix GitHub issues.

**Program Repair**: Automated program repair has had a long history, with many benchmarks covering different scopes and languages. These include DroixBench (Tan et al., 2018) for android apps; Defects4J (Just et al., 2014), GitBug-Java (Silva et al., 2024b), and growingBugs (Jiang et al., 2021; 2022a;b) for real-world Java; Bugsinpy (Widyasari et al., 2020) for Python; BugSwarm (Tomassi et al., 2019) for multilingual; DebugBench (Hu et al., 2024), LiveCodeBench (Jain et al., 2024b), and Codeflaws (Tan et al., 2017) for LeetCode-style problems; and many more.

Historically, there have been many techniques for this task, including heuristic-based APR (using genetic programming to explore the search space of the correct patch), constraint-based APR (treating repair as a constraint-solving task), pattern-based APR (apply expert hand-crafted repair templates), and learning-based APR (using language models) (Zhang et al., 2024a). More recently, with LLMs, there have been agent-based approaches such as FixAgent (Lee et al., 2024) using agents specializing in different aspects of debugging, and RepairAgent (Bouzenia et al., 2024) that invokes suitable tools. On the other hand, Agentless (Xia et al., 2024a) uses a three-phase process of localization, repair, and patch validation.

Finally, program repair has also been used as a tool to improve code generation, where error messages and incorrect test cases are fed back into the model to improve code generation (Madaan et al., 2023; Chen et al., 2024; Zhang et al., 2023a; Olausson et al., 2024; Zhong et al., 2024a; Tang et al., 2025). This is also known as self-repair or self-debugging. For a much more comprehensive survey of automated program repair, we recommend the reader check out this website[9].

**Code Understanding and Question Answering**: Code understanding with language models has been studied for many years. In earlier days, researchers used the CodeXGLUE (Lu et al., 2021) benchmark containing tasks such as clone detection, code search, code summarization, and so on. Nam et al. (2024) create an IDE plugin containing features that help users understand code through explaining highlighted sections of code and explaining domain-specific code. Yang et al. (2025) present a survey touching on reasoning-enhanced code intelligence.

### H.5. Scaffolding and Meta-Code

Beyond code generation, the broader software engineering ecosystem includes DevOps workflows, CI/CD pipelines, and Infrastructure-as-Code (IaC). LLMs have shown particular promise in generating, debugging, and explaining CI/CD configurations (e.g., GitHub Actions, Jenkinsfiles), assisting with environment setup, test orchestration, and deployment logic. A case study at Ericsson (Chaudhary et al., 2024) demonstrates how an LLM-based chatbot can support CI/CD question answering, enabling engineers to better understand and manage deployment pipelines. LLMs are also being explored for automated testing across heterogeneous software environments. ExecutionAgent (Bouzenia & Pradel, 2024) presents a language model-driven agent that autonomously installs, configures, and runs test suites for arbitrary projects.

Beyond CI/CD and testing, LLMs are increasingly used to reason about configuration logic and scaffolding code, which is a critical but often overlooked layer of modern software systems. For instance, Yin et al. (2011) conducted an empirical study of real-world configuration errors, identifying systemic causes of failure such as external dependencies, inter-parameter violations, and overlooked default parameters. Building on this line of work, Ciri (Lian et al., 2024) confirms the feasibility of using LLMs for configuration validation. Further, in the domain of IaC, an empirical study of 812 open-source Terraform projects found that while access policies are commonly adopted, critical practices like encryption at rest are often neglected (Verdet et al., 2023). This highlights the opportunity for LLMs to assist practitioners in detecting and enforcing security best practices in IaC configurations.

### H.6. Formal Verification

There are a variety of programming languages designed with different principles to support formal verification. Some of the popular ones include TLA (Lamport, 1994), Coq (The Coq Development Team, 2024), Lean (De Moura et al., 2015), Dafny (Leino, 2010), Isabelle (Nipkow et al., 2002), and Verus (Lattuada et al., 2024).

---

[9]https://program-repair.org/

Formal software verification has seen a few great successes in the last few years: Astrée (Cousot et al., 2005) was able to completely verify that Airbus A340's primary flight-control software had no run-time errors, verifying 132,000 lines of C code. More recently, formal methods have been applied to verify a cryptographic server (Erbsen et al., 2024) and an IoT lightbulb at both a hardware and software level (Erbsen et al., 2021). CompCert (Leroy et al., 2016), a verified compiler and seL4 (Klein et al., 2009), a verified microkernel are demonstrations that formal methods could be promising for verifiable code. At Amazon, formal methods been used to verify and protect cryptographic software (Goel et al., 2024), cloud resources (Xu et al., 2024b), and authorization (Disselkoen et al., 2025). Notably, SV-COMP (Beyer, 2023) is an annual competition designed to evaluate program verifiers using a curated benchmark of verifiable C and Java code. It even includes samples from the Linux Driver Verification (LDV) project (Beyer & Petrenko, 2012), aiding the verification of Linux kernel device drivers. For more applications, we refer the reader to the survey in Huang et al. (2023).

Recently, the ability of LLMs to write formal verification code. Benchmarks like DafnyBench (Loughridge et al., 2024) and miniCodeProps (Lohn & Welleck, 2024) were designed to measure the ability of LLMs to write software proofs in Dafny and Lean, respectively. In Dafny, Poesia et al. (2024) use a combination of search and prompting to create a synthetic dataset of annotations greatly improving performance on DafnyBench. Clover (Sun et al., 2024a) generates code alongside consistency checks (like Dafny annotations), Li et al. (2025c) employ Dafny as an intermediate language to improve code generation, and Misu et al. (2024) explore prompting and retrieval to generate Dafny. In Rust, Verus is a popular formal verification language, with AutoVerus (Yang et al., 2024a) and AlphaVerus (Aggarwal et al., 2024) generating verified specifications and proofs for Rust functions. There are also many IDE plugins designed to help humans to write code in formal languages such as Dafny and Lean such as Silva et al. (2024a), Lean Copilot (Song et al., 2024), and llmstep (Welleck & Saha, 2023).

Finally, there is a growing interest of work in using formal languages like Lean for mathematical theorem proving, which is covered comprehensively in Li et al. (2024f) and Yang et al. (2024d).

