# OpenReview forum: "Position: Future Research and Challenges Remain Towards AI for Software Engineering"
_ICML.cc/2025/Position_Paper_Track — ICML 2025 Position Paper Track poster_

### Official Review · Reviewer_hndP · 2025-02-19

**Significance:** 4
**Argument Clarity:** 3
**Rating:** 4
**Confidence:** 5

**Questions:**

The authors may consider different code intelligent scenarios. For example, in some non-critical scenarios, the generated code can be run and obtained feedback. But in other critical scenarios, the code cannot be run due to safety concerns, etc.

**Discussion Potential:**

3

**Paper Summary:**

This position paper discussed AI for software engineering. It provides a taxonomy of tasks and measures and outlines 10 major bottlenecks. It also gives possible paths towards future research.

## Update after rebuttal
I appreciate the author response.

**Position:**

Yes

**Position In Title:**

Yes

**Related Work:**

4

**Strengths And Weaknesses:**

Strengths:
- The paper is crystal clear and very easy to understand. The organization of the content is excellent.
- This position paper covers many typical aspects of AI4SE. I believe it is a good overview to summarize the current status.
- Several technical spots are very interesting, such as constrained decoding, controllable forgetting, etc.

Weaknesses:
- The paper writing seems a little rushed. There are some grammatical issues scattered in the paper:
  - Page 3, line 143: can easily (be) fixed
  - Page 4, line 174: Sec. 3
  - Page 7, line 360: describing properties (of) the code
- There are many abbreviations in the paper and they are not friendly to readers outside the software engineering community. The authors should give their full names as many as possible. For example, until page 8, I finally understand what is CI/CD.
- The authors may consider to show more empirical (and representative) results in the paper. I think page 6, Qwen on HumanEval is a good example. But I hope to see more. (This is just a suggestion, not a weakness)

**Support:**

3

---

> ### Author Rebuttal · Authors · 2025-03-31
>
> **Interesting technical spots (e.g. constrained decoding, controllable forgetting)**
>
> We appreciate the interest! In addition to the current proposed future technical directions, we will plan to add more ideas based on further feedback with researchers and industry players in the field since the submission deadline (see response to Reviewer HqMt).
>
> **More empirical results**
>
> Following up on the issues that we have mentioned in our position paper, we have been compiling a collection of examples and case studies that provide evidence of the challenges we mention. We will briefly overview them below and include all of them in the Appendix of our final paper. The full list of case studies is given in the response to Reviewer HqMt.
>
> **Different code intelligence scenarios**
>
> This is a great point that is closely related to the challenges in Sec 3.3 and 3.4, falling under “identifying when human input is necessary” and “implicit constraints”. We believe LLMs must learn to infer implicit constraints and seek clarification from the human on each scenario if the constraints are unclear. We will include this under the “Training models to collaborate with humans” section listed in our response to Reviewer HqMt.

---

### Official Review · Reviewer_HqMt · 2025-03-08

**Significance:** 4
**Argument Clarity:** 3
**Rating:** 4
**Confidence:** 4

**Questions:**

Please refer to the weaknesses.

**Discussion Potential:**

3

**Paper Summary:**

This paper provides a thorough view of the key task, critical challenges, and promising directions towards AI for software engineering. The promising research directions could alleviate these challenges and advance AI towards being a more capable software engineer.

**Position:**

Yes

**Position In Title:**

Yes

**Related Work:**

3

**Strengths And Weaknesses:**

Strengths:
1. The paper identifies six key tasks of AI for software engineering.
2. The paper highlights ten critical challenges in software engineering tasks.
3. The paper provides valuable insights into potential future research paths.


Weaknesses:
1. The paper lacks concrete empirical results or case studies that demonstrate how the discussed challenges and tasks manifest in real-world applications. Including practical examples or results from current AI-driven SWE tools would strengthen its argument.
2. Some of the proposed research directions, such as developing a "world model" for code or improving human-AI collaboration, are introduced but not explored in great depth.

**Support:**

3

---

> ### Author Rebuttal · Authors · 2025-03-31
>
> **Lack of Empirical Results**
>
> Following up on this paper, we have been compiling case studies with evidence of the challenges in Sec 3. Here, we give one example and overview several more that will be included in the Appendix of the revision.
>
> >Example: Library Usage Failure in Lean (Sec 3.8). Here, the LLM hallucinates a theorem in the low-resource language Lean. The LLM makes up a theorem named `infinitelyManyPrimes_arithmeticSequence`. While the model understands that it should use Dirichlet's theorem, the actual theorem is named `forall_exists_prime_gt_and_eq_mod`. This theorem occurs very few times in all of the currently available Lean code on GitHub, so it is likely that LLMs did not train on it many times.
> >```
> theorem infinitely_many_primes_of_form_6k_plus_1 :
>   ∀ n : ℕ, ∃ p : ℕ, p > n ∧ Prime p ∧ ∃ k : ℕ, p = 6 * k + 1 := by
>   intro n
>   -- Use Dirichlet's theorem for arithmetic progressions
>   have h := Nat.infinitelyManyPrimes_arithmeticSequence 6 1
>
> Additional case studies and failure modes are listed below (about half, due to rebuttal length limit):
> - 2.2: React Fiber (performance-motivated major refactoring in React), Google Chrome (V8 achieving a 20x performance improvement through cross-layer optimization)
> - 2.4: OSS-Fuzz (detecting a heap-buffer-overflow on FreeType), Project Zero (root cause and variance analysis)
> - 2.5: Ciri (config validation), Terrateam (distinguishing permission levels)
> - 2.6: Ariane 5 and Therac-25 (unverified software disasters), Coverity (false positives in property verification)
> - 3.3: In astropy-#14181 of SWE-Bench, the issue requests support for a new input file format but is vague. While both “read” and “write” methods are desired, model code only contains a “read” method.
> - 3.4: LLMs generate an incomplete postcondition when verifying a ring buffer implementation in Verus
> - 3.6: Models failing to localize errors when looking at JSON payload logs in Datadog, failure to retrieve relevant files in SWE-bench task “chartjs_Chart.js-7951” leads to incorrect generation
> - 3.8: Gemma3-27B uses invalid indexing notation when generating Triton code for dot product
> - 3.9: models use Lean 3 syntax when generating Lean 4 proofs, defaulting to Python 3.5 “typing.List” module syntax after Python 3.9 enabled direct use of built-in types
> - 3.10: challenge of developing a formal logic to verify file crashes and recoveries
>
> **No Human-AI in Sec 4**
>
> We will add a new section about HAI, including:
> 1. Human-Centric Data Curation
> - Collecting data in real-world scenarios with user-provided vague specifications, incomplete requirements, visual sketches for UIs, and multi-turn generation
> 2. Training models to collaborate
> - Using specifications beyond natural language: formal specifications and tests to clearly describe intent
> - Quantifying uncertainty and communicating proactively: training models to know when to communicate with the user (pragmatically), asking for clarification in ambiguity, acknowledging limitations and quantifying uncertainty
> 3. Scaffolding human supervision: making it easy for humans to make informed decisions about whether to accept AI-written code
> - Summarizing existing code, interactive verification (e.g. showing runtime states), presenting informative test cases to the user
>
> **Lack of Depth in Proposed Directions**
>
> We will add more depth to the revised version and include ideas such as the following:
>
> 4.1
> - Scaling up collection of executable codebases for RL
>
> 4.2
> - Augmenting data to increase code understanding: Including control flow graphs of programs, annotating code with variable types, reachability and liveness analyses, training with program states
> - Curating granular data: capturing real-time interactions during the development process: collecting data with user coding history such as code-edits, build outcomes, code reviews, and changes to a repository over time.
> - Sourcing for diverse SWE tasks: using existing GitHub code to curate prompts for other tasks (e.g. changes over time for program repair and code optimization)
>
> 4.3
> - Taking software anti-patterns (e.g. CWE vulnerabilities such as buffer overflow patterns) and steering models away from them by maximizing their NLL
>
> 4.4
> - Neurosymbolic methods: incorporating abstract interpretation, concolic testing, model checking, and type-checking with LLMs
> - Using deductive synthesis approaches to generate code in intermediate languages which can be manipulated symbolically to preserve correctness guarantees
>
> 4.5
> - Keeping a memory bank of code-specific information for each repository in order to enable “on-the-fly” code navigation (such as retrieving the right files via traversing the file directory)
> - Jointly training retrieval models and generation models for more semantic embeddings
> - Prefix tuning in specialized contexts: training context-specific embeddings depending on style and version constraints
> - Learning on the fly: getting up to speed with only a few samples (e.g. DreamCoder)

---

> > ### Comment · Reviewer_HqMt · 2025-04-07
> >
> > Thanks for the clarifications. I have raised the score.

---

### Official Review · Reviewer_fWo9 · 2025-03-13

**Significance:** 3
**Argument Clarity:** 3
**Rating:** 4
**Confidence:** 4

**Questions:**

1) What is the stated position?

**Discussion Potential:**

3

**Paper Summary:**

This paper provides a high level overview of progress in AI for software engineering. First a comprehensive taxonomy of different tasks is presented along with examples of each. Different tasks are evaluated based on scope, algorithmic complexity, and level of human intervention. Tasks are divided into high level groups including code generation and transformation. Several different challenges are described as well as high-level suggestions for tackling those challenges.

**Position:**

No

**Position In Title:**

No

**Related Work:**

3

**Strengths And Weaknesses:**

Strengths:
- The taxonomy and different metrics  are a useful way of comparing different works.

Weaknesses:
- This paper is ultimately a survey paper, not a position paper. There is no stated position nor evidence for that position either.
- Under the “Alternative views” section (line 400), it is implied that the stated position is that AI for software engineering has lots of room for improvement. Such a statement is a) vague and b) not something that people disagree on. The authors claim that in industry it is believed that only limited progress is needed but there is no evidence for this. It seems unlikely that most people would disagree with the statement.
-  Similar to the other weaknesses, most of the challenges are fairly common to most tasks in the LLM era. The authors are not really stating anything new or than a typical ‘laundry list’ of problems with LLMs (with some specific to code generation).

As a survey paper, this paper is decent and provides a nice way to compare different works. However, given that this is for a position paper and there is no position, I recommend rejection. The position should also be something more descriptive than “AI for software engineering needs to be improved.” Ideally there is some insight or discussion points in a position paper, but this work has none.

**Support:**

3

---

> ### Author Rebuttal · Authors · 2025-03-31
>
> **Lack of clear position**
>
> Thanks a lot for the review! We apologize for any potential misunderstandings about the position posed in our paper. We hope that the clarification below addresses the reviewer’s concerns, and welcome more suggestions about how to make our position clearer.
>
> One popular view (an “alternative view”) is that scaling to bigger models, writing better prompts, and embracing agent paradigms will be sufficient for resolving the majority of today’s challenges in AI for code. This is not an uncommon position: many believe that pursuing the research directions we suggest are worthless, but rather that what we need is larger models and more coding data. There is much discourse about scaling getting us towards automating SWE, and the authors collectively know many people (primarily in industry) who hold this view.
>
> For example, Anthropic CEO Dario Amodei said AI will write 90% of SWE code within 3-6 months and every line of code within the next year [1]. Mark Zuckerberg says AI will replace mid-level engineers by 2025 [2]. Salesforce CEO Marc Benioff says Salesforce won’t hire engineers due to AI [3]. Sam Altman says that “the big thing will come with agentic coding,” saying that “we just need a little longer”. We take a more pessimistic view, believing that these challenges will not just go away with agentic coding, but rather require research directions like the ones we mention.
>
> Our position is in direct opposition to this: rather, that there is quite a bit of research work left to be done before achieving highly automated degrees of SWE. To support this view, we describe 1) challenges we believe must be resolved in order to reach a high level of automation in SWE (Sec. 3) and 2) an opinionated take on research directions that are necessary based on the challenges we pose.
>
> Finally, as structurally similar papers like [5] were historically accepted to the ICML Position Track, we believe our paper lies in scope of this call. We hope that the reviewer can reconsider the paper's fit in light of our clarification, and we look forward to incorporating any further suggestions that can help make things clearer.
>
> [1] https://www.entrepreneur.com/business-news/anthropic-ceo-predicts-ai-will-take-over-coding-in-12-months/488533
>
> [2] https://www.forbes.com/sites/quickerbettertech/2025/01/26/business-tech-news-zuckerberg-says-ai-will-replace-mid-level-engineers-soon/
>
> [3] https://sfstandard.com/2025/02/27/salesforce-marcbenioff-layoffs-tech-agents/
>
> [4] https://www.businessinsider.com/sam-altman-ai-coding-job-market-software-engineer-students-openai-2025-3
>
> [5] Morris, et al, 2025. Future Directions in the Theory of Graph Machine Learning
>
> **Challenges are general, nothing new stated**
>
> We already have many code-specific challenges and future work, including:
> - 3.2: integrating LLMs with SWE tools (L188-205), which is very underexplored
> - 3.4: the four points mentioned in this section are all code-specific
> - 3.5: while planning is a general issue for LLMs, developing abstractions, library learning, and choosing data representation are code specific
> - 3.8/3.9: while these can be seen as LLM challenges, they are crucial in code due to proprietary corporate codebases and changing APIs
> - 4.1: using symbolic tools to generate verifiable synthetic data, sampling programs from DSLs (L330-345)
> - 4.2: using PL techniques to add symbolic information about programs before training (L354-367), augmenting training with execution information (L373-374)
> - 4.3: integrating AI into the CI/CD cycle
> - 4.4: using language grammar to do constrained decoding, using AST structures in training (L417-424)
>
> In order to make our paper more insightful, we plan to add more case studies/examples of the challenges in Sec. 3 and go into more depth with our future directions in Sec. 4 (listed in our response to Reviewer HqMt). At NeurIPS 2024, we found that many researchers thought AI4Code was an industry-dominated field with little potential research ideas remaining. Our hope is to give ML researchers a fresh perspective on the field and new ideas by highlighting open unresolved challenges. Many of these challenges and future directions call upon ideas from programming languages and software engineering researchers less commonly known to ML folks.
>
> Finally, we believe our paper will elicit discussion when it comes to whether the directions in Sec 4 are the correct ways to tackle the challenges in Sec 3. Many researchers will disagree with the directions we propose (e.g. many researchers believe semantic code embeddings will arise naturally).

---

> > ### Comment · Reviewer_fWo9 · 2025-04-01
> >
> > Thank you for all of the clarifications as well as pointing to the similar position paper. I think that putting some of these quotes  at the beginning would help. One way to frame it is that researchers believe this but industry believes this.
> >
> > I have updated the score to 4.

---

### Official Review · Reviewer_eh11 · 2025-03-14

**Significance:** 3
**Argument Clarity:** 3
**Rating:** 3
**Confidence:** 4

**Questions:**

1. Why, in particular, does SWE-Bench (and related benchmarks) fail to represent real-world use cases? What evidence do you have around that?
2. Why do you remain skeptical of long-horizon code planning? Can you share more related work or evidence for why this is the case?
3. You mention that "In addition, code embeddings 281 generally group code together via syntactic similarity (Ma 282 et al., 2024; Utpala et al., 2023), which can make it hard to 283 retrieve crucial snippets that are semantically relevant but 284 syntactically unrelated.” Is this true? Can you provide more discussion as to why this would necessarily exclude semantically relevant queries -- aren't embeddings often semantic as well?

**Discussion Potential:**

3

**Paper Summary:**

The position is: more research efforts are needed towards AI for software engineering. The paper describes a taxonomy of tasks and measures; main bottlenecks; and paths towards making progress towards the bottlenecks of the research.

**Position:**

Yes

**Position In Title:**

No

**Related Work:**

3

**Strengths And Weaknesses:**

Strengths:
- Clear and wide-ranging taxonomy presented
- Good recommendation to create more benchmarks for real-world use cases

Weaknesses:
- Taxonomy does not include vulnerability detection / vulnerability fixing
- Focuses largely on research from academia but doesn't talk as much about AI products / companies. Does not mention tools such as Cursor / Claude Code, which might be worth talking about given their widespread use for AI SWE and their research innovations in their own right.
- Position could be stated more strongly. And just saying "more research in AI SWE is needed" could probably be made sharper, such as "we need more benchmarks for real-world use cases"

**Support:**

2

---

> ### Author Rebuttal · Authors · 2025-03-31
>
> **Omission of vulnerability detection / vulnerability fixing**
>
> We agree this is a very important topic and will add the following:
> - Vulnerability detection: identifying exploitable flaws compromising security, stability, or correctness (IRIS, LLMDFA, InferROI, BigSleep)
> - Software testing: unit tests, property-based tests, code coverage, evaluating system behavior across complex interactions or multiple components (Fuzz4All, KernelGPT, OSS-Fuzz)
> - Vulnerability fixing: automatic program repair, heuristic-based, constraint-based (repair as a SMT task)
>
> **Omission of AI products / companies**
>
> Our challenges and research directions apply generally to both academia and industrial AI tools. We have also consulted people working on AI for code in industry (e.g. Cognition AI, Mistral, DeepSeek, Cohere, Anthropic, Meta, Google). Sec 3.3 and 3.4 applies directly to the user experience when using industry tools. In Sec 4, we will add a new subsection on Human-AI collaboration to expand on this, including:
> - Tools like Cursor can use fine-grained user data to personalize models and predict user behavior
> - AI coding assistants proactively communicating to humans when specifications are unclear and when they feel unconfident in their response
>
> **Stronger Position**
>
> Our position consists of two aspects: 1) challenges we believe must be resolved in order to reach a high level of automation in SWE (Sec. 3) and 2) an opinionated take on fruitful research directions that should be pursued based on 1). We will add a stronger position statement for each subsection in Sec. 3. For example, in 3.1, we will say “Today's code LLM evaluations focus on a narrow set of tasks, suffer from potential contamination, and do not reliably measure real-world software engineering abilities."
>
> **SWE-Bench vs. Real World**
>
> We discuss 3 issues with SWE-Bench:
> 1. Task Diversity and Capability Isolation: Relying solely on end-to-end code generation evaluations makes it difficult to discern performance on other tasks like debugging, refactoring, and optimization (Sec 2). For projects at larger scopes (e.g. designing an end-to-end framework), these capabilities will inevitably be necessary as it will be impossible to generate correct code in one go.
> 2. Data Contamination and Data Quality: [1] show that SWE-Bench suffers greatly from contamination (older issues are fixed much more frequently than newer issues), solution leakage (many solutions were provided in the comments), and weak test cases (31% of the passed patches are suspicious due to weak test cases).
> 3. Construct Validity (L177-184): Things like multi-turn code generation, designing user interfaces, responding to user questions/comments, and writing clean and idiomatic code are difficult to measure with automatic unit testing. Engineers at Cognition AI have also told us that SWE-Bench provides a poor signal of real world utility.
>
> [1] https://arxiv.org/abs/2410.06992
>
> **Skepticism about code planning**
>
> Long-horizon planning is difficult for several reasons. An instructive example is to consider whether code LLMs will be able to design a framework for training and evaluating LLMs (akin to HuggingFace’s Trainer, vLLM, or SGLang) from scratch without seeing them in the training data. We believe the answer is negative. First, there are many design decisions that need to be made, such as how to best structure the classes for data, model, post-training techniques (see [1]). Second, the framework also needs to be extensible and quickly adapt to new model releases. Finally, this is a relatively out-of-distribution example with a challenge of gathering data on the evolution of a codebase and its structure over time.
>
> Also, LLM-written code may be hard to maintain, often lacking modularity and having poor code quality. One reason is that RL-style approaches only reward correct solutions with no regard for quality. Empirically, LLM written solutions are often more complex than human-written counterparts. [2] identify that library or tool reuse can be difficult in coding and formal math, making code less modular. [3] find LLMs prefer to repeat existing code instead of making use of existing abstractions.
>
> [1] https://docs.vllm.ai/en/latest/design/arch_overview.html
>
> [2] https://arxiv.org/abs/2410.20274
>
> [3] https://r2e.dev/
>
> **Semantic nature of code embeddings**
>
> While code embeddings do contain some semantic information, they struggle to retrieve semantically equivalent code. Figure 2(f) in [1] shows that embeddings of semantically equivalent code in different languages are further apart than that of semantically different code in the same language. [2] test code embedding models on syntactic and semantic probing tasks, finding high variability. Results on benchmarks such that CrossCodeEval and BRIGHT show that models struggle to find the relevant context when semantic relevance and code reasoning are desired.
>
> [1] https://arxiv.org/pdf/2310.16803
>
> [2] https://openreview.net/pdf?id=L0EnFg4Fev

---

### Decision · Program_Chairs · 2025-04-30

**Decision:**

Accept (poster)

**Comment:**

Very important area.
Agreement by reviewers.
Taxonomy and challenges are good.
Position could be stated more strongly.
Cover more of the commercial tools, Claude etc.

Authors should revise the title to state the position, as instructed in the 2025 CFP (which has more specific guidance on titles than the 2024 CFP).  It may help to find a verb to include.